# Offline Data Enhanced On-Policy Policy Gradient with Provable Guarantees

**Yifei Zhou** [*]
University of California, Berkeley
{yifei_zhou@berkeley.edu

**Ayush Sekhari** [*]
MIT
sekhari@mit.edu

**Yuda Song**
Carnegie Mellon University
yudas@cs.cmu.edu

**Wen Sun**
Cornell University
ws455@cornell.edu

## Abstract

Hybrid RL is the setting where an RL agent has access to both offline data and online data by interacting with the real-world environment. In this work, we propose a new hybrid RL algorithm that combines an on-policy actor-critic method with offline data. On-policy methods such as policy gradient and natural policy gradient (NPG) have shown to be more robust to model misspecification, though sometimes it may not be as sample efficient as methods that rely on off-policy learning. On the other hand, offline methods that depend on off-policy training often require strong assumptions in theory and are less stable to train in practice. Our new approach integrates a procedure of off-policy training on the offline data into an on-policy NPG framework. We show that our approach, in theory, can obtain a *best-of-both-worlds* type of result — it achieves the state-of-art theoretical guarantees of offline RL when offline RL-specific assumptions hold, while at the same time maintaining the theoretical guarantees of on-policy NPG regardless of the offline RL assumptions' validity. Experimentally, in challenging rich-observation environments, we show that our approach outperforms a state-of-the-art hybrid RL baseline which only relies on off-policy policy optimization, demonstrating the empirical benefit of combining on-policy and off-policy learning. Our code is publicly available at https://github.com/YifeiZhou02/HNPG.

## 1 Introduction

On-policy RL methods, such as direct policy gradient (PG) methods (Williams, 1992; Sutton et al., 1999; Konda & Tsitsiklis, 1999; Kakade, 2001), are a class of successful RL algorithms due to their compatibility with rich function approximation (Schulman et al., 2015), their ability to directly optimize the cost functions of interests, and their robustness to model-misspecification (Agarwal et al., 2020). While there are many impressive applications of on-policy PG methods in high-dimensional dexterous manipulation (Akkaya et al., 2019), achieving human-level performance in large-scale games (Vinyals et al., 2019), and finetuning large language model with human feedback (Ouyang et al., 2022), the usage of on-policy PG methods is often limited to the setting where one can afford a huge amount of training data. This is majorly due to the fact that on-policy PG methods do not reuse old data (i.e., historical data that are not collected with the current policy to optimize or evaluate).

On the other hand, offline RL asks the question of how to reuse existing data. There are many real-world applications where we have pre-collected offline data (Fan et al., 2022; Grauman et al., 2022), and the goal of offline RL is to learn a high-quality policy purely from offline data. Since offline data typically is generated from sub-optimal policies, offline RL methods rely on off-policy learning (e.g., Bellman-backup-based learning such as Q-learning and Fitted Q Iteration (FQI) (Munos & Szepesvári, 2008)). While the vision of offline RL is promising, making offline RL work in both theory and practice is often challenging. In theory, offline RL methods rely on strong assumptions on the function

---

[*]First two authors contribute equally.

approximation (e.g., classic off-policy Temporal Difference (TD) Learning algorithms can diverge without strong assumptions such as Bellman completeness (Tsitsiklis & Van Roy, 1996)). In practice, unlike on-policy PG method which directly performs gradient ascent on the objective of interests, training Bellman-backup based value learning procedure in an off-policy fashion can be unstable (Kumar et al., 2019) and less robust to model misspecification (Agarwal et al., 2020). In this work, we ask the following question: **Can we design an RL algorithm that can achieve the strengths of both on-policy and offline RL methods?** We study this question and provide an affirmative answer under the setting of hybrid RL (Ross & Bagnell, 2012; Song et al., 2023), which considers the situation where in addition to some offline data, the learner can also perform online interactions with the underlying environment to collect fresh data. Prior hybrid RL works focus on the simple approach of mixing both offline data and online data followed by iteratively running off-policy learning algorithms such as FQI (Song et al., 2023) or Soft Actor-Critic (SAC) (Nakamoto et al., 2023; Ball et al., 2023)—both of which are off-policy methods that rely on Bellman backup or TD to learn value functions from off-policy data. We take an alternative approach here by augmenting on-policy PG methods with an off-policy learning procedure on the given offline data. Different from prior work, our new approach combines on-policy learning and off-policy learning, thus achieving the best of both worlds guarantee. More specifically, on the algorithmic side, we integrate the Fitted Policy Evaluation procedure (Antos et al., 2007) (an off-policy algorithm) into the Natural Policy Gradient (NPG) (Kakade, 2001) algorithm (an on-policy framework). On the theoretical side, we show that when standard assumptions related to offline RL hold, our approach indeed achieves similar theoretical guarantees that can be obtained by state-of-art theoretical offline RL methods which rely on pessimism or conservatives (Xie et al., 2021), while at the same time, our approach always maintains the theoretical guarantee of the on-policy NPG algorithm, regardless of the validity of the offline RL specific assumptions. Thus, our approach can still recover the on-policy result while the offline component fails.

On the practical side, we verify our approach on the challenging rich-observation combination lock problem (Misra et al., 2020) where the agent has to always take the only correct action at each state to get the final optimal reward (see Section 6 for more details). This RL environment has been extensively used in prior works to evaluate an RL algorithm's ability to do representation learning and exploration simultaneously (Zhang et al., 2022b; Song et al., 2023; Agarwal et al., 2023). Besides the standard rich-observation combination lock example, we propose a more challenging variant where the observation is made of real-world images from the Cifar100 dataset (Krizhevsky, 2009). In the Cifar100 augmented combination lock setting, the RL agent can only access images from the training set during training and will be tested using images from the test set. Unlike standard Mujoco environments where the transition is deterministic and initial state distribution is narrow, our new setup here stresses testing the generalization ability of an RL algorithm when facing real-world images as states. Empirically, on both benchmarks, our approach significantly outperforms baselines such as pure on-policy method PPO and a hybrid RL approach RLPD (Ball et al., 2023) which relies on only off-policy learning.

## 2 RELATED WORKS

**On-policy RL.** On-policy RL defines the algorithms that perform policy improvement or evaluation using the current policy's actions or trajectories. The most notable on-policy methods are the family of direct policy gradient methods, such as REINFORCE (Williams, 1992), Natural Policy Gradient (NPG) (Kakade, 2001), and more recent ones equipped with neural network function approximation such as Trust Region Policy Optimization (TRPO) (Schulman et al., 2015) and Proximal Policy Optimization (PPO) (Schulman et al., 2017). In general, on-policy methods have some obvious advantages: they directly optimize the objective of interest and they are nicely compatible with general function approximation, which contributes to their success on larger-scale applications (Vinyals et al., 2019; Berner et al., 2019). In addition, (Agarwal et al., 2020) demonstrates the provable robustness to the "Delusional Bias" (Lu et al., 2018) while only part of the model is well-specified.

**Off-policy / offline RL.** Off-policy learning uses data from behavior policy that is not necessarily the current policy that we are estimating or optimizing. Since off-policy methods rely on the idea of bootstrapping (either from the current or the target function), in theory, stronger assumptions are required for successful learning. Foster et al. (2021) showed that realizability alone does not guarantee sample efficient offline RL (in fact, the lower bound could be arbitrarily large depending on the state space size). Stronger conditions such as Bellman completeness are required. It is also well-known that in off-policy setting, when equipped with function approximation, classic TD algorithms indeed do not guarantee to converge (Tsitsiklis & Van Roy, 1996), and even converged, the fixed point

solution of TD can be arbitrarily bad (Scherrer, 2010). In addition, (Agarwal et al., 2020) provided counter-examples for TD/Q-learning style algorithms' failures on partially misspecified models. These negative results all indicate the challenges of learning with offline or off-policy data. On the other hand, positive results are present when stronger assumptions such as Bellman completeness (Munos & Szepesvári, 2008). While these assumptions are off-policy/offline learning specific and can be strong, the fact that TD can succeed in practice implies such conditions can hold in practice (or at least hold approximately).

**Hybrid RL.** Hybrid RL (Song et al., 2023) defines the setting where the learning agent has access to both offline dataset and online interaction with the environment. Previous hybrid RL methods (Song et al., 2023; Ross & Bagnell, 2012; Ball et al., 2023; Nakamoto et al., 2023) perform off-policy learning (or model-based learning) on the dataset mixed with online and offline data. In particular, HyQ (Song et al., 2023) provides theoretical justification for the off-policy approach, and the guarantees presented by HyQ still require standard offline RL conditions to hold (due to the bootstrap requirement). However, our new approach is fundamentally different in algorithm design: although our offline component is still inevitably off-policy with the ideas of bootstrap, we perform on-policy learning with the data collected during online interaction without bootstrap, which gives us a doubly robust result when the offline learning specific assumptions (e.g., Bellman completeness) do not hold. Additionally, we would like to mention that some other works (Gu et al., 2017b;a; Xiao et al., 2023; Zhao et al., 2023; Lee et al., 2021), also explored the possibility of achieving the best of both worlds of on-policy and off-policy learning. Despite achieving empirical success, their theoretical guarantees still require a strong coverage condition on the reset distribution, while this work presents a doubly-robust guarantee when either the offline or the on-policy condition holds.

## 3 PRELIMINARIES

We consider discounted infinite horizon MDP $\mathcal{M} = \{\mathcal{S}, \mathcal{A}, \gamma, r, \mu_0, P\}$ where $\mathcal{S}, \mathcal{A}$ are state-action spaces, $\gamma \in (0, 1)$ is the discount factor, $r(s, a) \in [0, 1]$ is the reward, $\mu_0 \in \Delta(\mathcal{S} \times \mathcal{A})$ is the initial reset distribution over states and actions (i.e., we reset based on a state and action sampled from $\mu_0$), and $P \in \mathcal{S} \times \mathcal{A} \mapsto \Delta(\mathcal{S})$ is the transition kernel. Note that assuming the reset distribution over the joint space $\mathcal{S} \times \mathcal{A}$, contrary to just resetting over $\mathcal{S}$, is a standard assumption used in policy optimization literature such as CPI and NPG (Kakade & Langford, 2002; Agarwal et al., 2021).

As usual, given a policy $\pi \in \mathcal{S} \mapsto \Delta(\mathcal{A})$, we denote $Q^\pi(s, a)$ as the Q function of $\pi$, and $V^\pi(s)$ as the value function of $\pi$. We denote $d^\pi \in \Delta(\mathcal{S} \times \mathcal{A})$ as the average state-action occupancy measure of policy $\pi$. We denote $V^\pi = \mathbb{E}_{s_0 \sim \mu_0} V^\pi(s_0)$ as the expected total discounted reward of $\pi$. We denote $\mathcal{T}^\pi$ as the Bellman operator associated with $\pi$, i.e., given a function $f \in \mathcal{S} \times \mathcal{A} \mapsto \mathbb{R}$, we have

$$\mathcal{T}^\pi f(s, a) = r(s, a) + \gamma \mathbb{E}_{s' \sim P(s,a), a' \sim \pi(s')}[f(s', a')].$$

In the hybrid RL setting, we assume that the learner has access to an offline data distribution $\nu$, from which it can draw i.i.d. samples $s, a \sim \nu, r = r(s, a), s' \sim P(s, a)$ to be used for learning (in addition to on-policy online interactions). The assumption that the learner has direct access to $\nu$ can be easily relaxed by instead giving the learner a dataset $\mathcal{D}$ of samples drawn i.i.d. from $\nu$. For a given policy $\pi$, we denote $d^\pi$ as the average state-action occupancy measure, starting from $\mu_0$.

In our algorithm, given a policy $\pi$, we will draw state-action pairs from the distribution $d^\pi$ defined such that $d^\pi(s, a) = (1 - \gamma)(\mu_0(s_1, a_1) + \sum_{t=1}^\infty \gamma^t \Pr^\pi(s_t = s, a_t = a))$, which can be done by sampling $h$ with probability proportional to $\gamma^h$, execute $\pi$ to $h$ starting from $(s_1, a_1) \sim \mu_0$, and return $(s_h, a_h)$. Given $(s, a), \pi$, to draw an unbiased estimate of the reward-to-go $Q^\pi(s, a)$, we can execute $\pi$ starting from $(s_0, a_0) := (s, a)$, every time step $h$, we terminate with probability $\gamma$ (otherwise move to $h + 1$); once terminated at $h$, return the sum of the *undiscounted* rewards $\sum_{\tau=0}^h r_\tau$. This is an unbiased estimate of $Q^\pi(s, a)$. Such kind of procedure is commonly used in on-policy PG methods, such as PG (Williams, 1992), NPG (Kakade, 2001; Agarwal et al., 2020), and CPI (Kakade & Langford, 2002). We refer readers to Algorithm 1 in Agarwal et al. (2021) for details.

## 4 HYBRID ACTOR-CRITIC

In this section, we present our main algorithm called Hybrid Actor-Critic (HAC), given in Algorithm 1. HAC takes as input the number of rounds $T$, a value function class $\mathcal{F}$, an offline data distribution $\nu$ (or equivalently an offline dataset sampled from $\nu$), and a weight parameter $\lambda$, among other parameters, and returns a policy $\hat{\pi}$. HAC runs for $T$ rounds, where it performs a few very simple steps at each round $t \in T$. At the beginning of every round, given a policy $\pi^t$ computed in the previous rounds, it first invokes the subroutine Hybrid Fitted Policy Evaluate (HPE), given in Algorithm 2,

---

**Algorithm 1** **H**ybrid **A**ctor-**C**ritic (HAC)

---

**Require:** Function class $\mathcal{F}$, offline data $\nu$, # of PG iteration $T$, HPE # of iterations $K_1, K_2$, weight parameter $\lambda$

1: Initialize $f^0 \in \mathcal{F}$, set $\pi_1(a|s) \propto \exp(f^0(s,a))$.
2: Set $\eta = (1-\gamma)\sqrt{\log(A)/T}$.
3: **for** $t = 1, \ldots, T$ **do**
4:     Let $f^t \leftarrow \text{HPE}(\pi^t, \mathcal{F}, K_1, K_2, \nu, \lambda)$.
5:     $\pi^{t+1}(a|s) \propto \pi^t(a|s) \exp(\eta f^t(s,a)), \qquad \forall s, a.$
6: **end for**
7: **Return** policy $\widehat{\pi} \sim \text{Uniform}(\{\pi_1, \ldots, \pi_{T+1}\})$.

---

**Algorithm 2** **H**ybrid **F**itted **P**olicy **E**valuation (HPE)

---

**Require:** Policy $\pi$, function class $\mathcal{F}$, offline distribution $\nu$, number of iterations $K_1, K_2$, weight $\lambda$

1: Initialize $f_0 \in \mathcal{F}$.
2: Sample $\mathcal{D}_{\text{on}} = \{(s, a, y = \widehat{Q}^\pi(s,a))\}$ of $m_{\text{on}}$ many on-policy samples using $\pi$.
3: Sample $\mathcal{D}_{\text{off}} = \{(s, a, s', r)\}$ of $m_{\text{off}}$ many offline samples from $\nu$.
4: **for** $k = 1, \ldots, K_1, \ldots, K_2$ **do**
5:     Solve the square loss regression problem to compute:

$$f_k \leftarrow \underset{f \in \mathcal{F}}{\text{argmin}} \; \widehat{\mathbb{E}}_{\mathcal{D}_{\text{off}}}(f(s,a) - r - \gamma f_{k-1}(s', \pi(s')))^2 + \lambda \widehat{\mathbb{E}}_{\mathcal{D}_{\text{on}}}(f(s,a) - y)^2. \quad (1)$$

6:     Sample fresh datasets $\mathcal{D}_{\text{off}}$ and $\mathcal{D}_{\text{on}}$ as in lines 2 and 3 above.
7: **end for**
8: **Return** $\bar{f} = \frac{1}{K_2 - K_1} \sum_{k=K_1+1}^{K_2} f_k$, and optionally $\mathcal{D}_{\text{off}}$ and $\mathcal{D}_{\text{on}}$.

---

to compute an approximation $f^t$ of the value function $Q^{\pi^t}$ corresponding to $\pi^t$. Then, using the function $f^t$, HAC computes the policy $\pi^{t+1}$ for the next round using the softmax policy update: $\pi^{t+1}(a \mid s) \propto \pi^t(a \mid s) \exp(\eta f^t(s,a))$ for $s \in \mathcal{S}$, where $\eta$ is the step size. This step ensures that the new policy does not change too much compared to the old policy.

We next describe the subroutine HPE, our key tool in the HAC algorithm. HPE algorithm takes as input a policy $\pi$, a value function class $\mathcal{F}$, an offline distribution $\nu$, and a weight parameter $\lambda$, among other parameters, and outputs a value function $f$ that approximates $Q^\pi$ of the input policy $\pi$. HPE performs $K_2$ many iterations, where on the $k$-th iteration, it computes a function $f_k$ based on the function $f_{k-1}$ from the previous rounds. At the $k$-th iteration, in order to compute $f_k$, HPE first collects a dataset $\mathcal{D}_{\text{on}}$ of $m_{\text{on}}$ many on-policy online samples from the input policy $\pi$, each of which consists of a triplet $(s, a, y)$ where $(s, a) \sim d^\pi$ and $y$ is a stochastic estimate for $Q^\pi(s,a)$ i.e. satisfies $\mathbb{E}[y] = Q^\pi(s,a)$ (e.g., $y$ can be obtained from a Monte-Carlo rollout). Then, HPE collects a dataset $\mathcal{D}_{\text{off}}$ of $m_{\text{off}}$ many offline samples $(s, a, s', r)$ from $\nu$, where $s' \sim P(\cdot \mid s, a)$ and $r \sim r(s,a)$. Finally, HPE computes the estimate $f_k$ by solving the optimization problem in (1). The first term in (1) corresponds to minimizing TD error with respect to $f_{k-1}$ under the offline data $\mathcal{D}_{\text{off}}$, and the second term corresponds to minimizing estimation error of $Q^\pi(s,a)$ under the online dataset $\mathcal{D}_{\text{on}}$. Note that the second term does not rely on the bootstrapping procedure. The relative weights of the terms is decided by the parameter $\lambda$, given as an input to the algorithm and chosen via hyperparameter tuning in our experiments. Typically, $\lambda \in [1, T]$. Finally, after repeating this for $K_2$ times, HPE outputs $\bar{f}$ which is computed by taking the average of $f_k$ produced in the last $K_2 - K_1$ iterations[1], where we ignore the first $K_1$ iterations to remove the bias due to the initial estimate $f_0$.

The key step in HPE is Eq. 1 which consists of an off-policy TD loss and an on-policy least square regression loss. When the standard offline RL condition — Bellman completeness, holds (i.e., the Bayes optimal $\mathcal{T}^\pi f_{k-1} \in \mathcal{F}$), HPE can return a function $\bar{f}$ that has the following two properties: (1) $\bar{f}$ is an accurate estimate of $Q^\pi(s,a)$ under $d^\pi$ thanks to the on-policy least square loss, (2) $\bar{f}$ has small Bellman residual under the $\nu^2$ — the offline distribution. On the other hand, without the Bellman completeness condition, due to the existence of the on-policy regression loss, we can still ensure

---

[1]Averaging is only needed for our theoretical guarantees. Our experiments in Section 6 do not perform averaging and use the last iterate.
[2]i.e. $\mathbb{E}_{s,a \sim \nu}(\bar{f}(s,a) - r(s,a) - \gamma \mathbb{E}_{s' \sim P(\cdot|s,a), a' \sim \pi} \bar{f}(s', a'))^2$

$\bar{f}$ is a good estimator of $Q^\pi$ under $d^\pi$. This property ensures that we always retain the theoretical guarantees of on-policy NPG. We illustrate these points in more detail in the analysis section.

**Parameterized policies.** Note that HAC uses softmax policy parameterization, which may be intractable when $\mathcal{A}$ is large (since we need to compute a partition function in order to sample from $\pi^t$). In order to circumvent this issue, we also consider a Hybrid Natural Policy Gradient (HNPG) algorithm (Algorithm 3), that directly works with a parameterized policy class $\Pi = \{\pi_\theta \mid \theta \in \Theta\}$, where $\theta$ is the parameter (e.g., $\pi_\theta$ can be a differentiable neural network based policy). The algorithm is very similar to HAC and runs for $T$ rounds, where at each round $t \leq T$, it first computes $f^t$, an approximation for $Q^{\pi_{\theta^t}}$, by invoking the HPE procedure. However, it relies on compatible function approximation to update the current policy $\pi_{\theta^t}$. For working with parameterized policies, HNPG relies on HPE to also supply an offline dataset $\mathcal{D}_{\text{off}}$ of $m_{\text{off}}$ many tuples $(s, a) \sim \nu$, and an on-policy online dataset $\mathcal{D}_{\text{on}}$ of $m_{\text{on}}$ many tuples $(s, a) \sim d^{\pi^t}$, which it uses to fit the linear critic $(w^t)^\top \nabla \ln \pi_{\theta^t}(a|s)$ in (2). We then update the current policy parameter via $\theta^{t+1} = \theta^t + \eta w^t$, similar to the classic NPG update for parameterized policy (Kakade, 2001; Agarwal et al., 2021) except that we fit the linear critic under both online and offline data. Another way to interpret this update rule is to investigate the form of $w^t$. Taking the gradient of the objective in (2) with respect to $w$, setting it to zero, and solving for $w$, we get that the stationary point should be in the form of $\left[\sum_{s,a}(\phi^t(s, a)(\phi^t(s, a))^\top)\right]^{-1} \sum_{s,a} \phi^t(s, a)\bar{f}^t(s, a)$. Using the fact that $\phi^t(s, a)$ is defined to be $\nabla \ln \pi_{\theta^t}(a|s)$, we see that $\sum_{s,a}(\phi^t(s, a)(\phi^t(s, a))^\top$ is *exactly the fisher information matrix computed using both online and offline data*. Thus our new approach extends the parameterized NPG (Kakade, 2001) to the hybrid RL setting in a principled manner.

---

**Algorithm 3** Hybrid NPG with Parameterized Policies (HNPG)

---

**Require:** Function class $\mathcal{F}$, PG iteration $T$, PE iterations $(K_1, K_2)$, offline data $\nu$, Params $\lambda, \eta$.
1: Initialize $f^0 \in \mathcal{F}$, and $\theta^1$ such that $\pi_{\theta^1} = \text{Uniform}(\mathcal{A})$.
2: **for** $t = 1, \ldots, T$ **do**
3:  $\quad f^t, \mathcal{D}_{\text{off}}, \mathcal{D}_{\text{on}} \leftarrow \text{HPE}(\pi^t, \mathcal{F}, K_1, K_2, \nu, \lambda)$.
4:  $\quad$ Let $\phi^t(s, a) = \nabla \log \pi_{\theta^t}(a|s)$ and $\bar{f}^t(s, a) = f^t(s, a) - \mathbb{E}_{a \sim \pi_{\theta^t}(s)}[f^t(s, a)]$.
5:  $\quad$ Solve the square loss regression problem to compute:
$$w^t \in \underset{w}{\operatorname{argmin}} \, \widehat{\mathbb{E}}_{\mathcal{D}_{\text{off}}}\left[(w^\top \phi^t(s, a) - \bar{f}^t(s, a))^2\right] + \lambda \widehat{\mathbb{E}}_{\mathcal{D}_{\text{on}}}\left[(w^\top \phi^t(s, a) - \bar{f}^t(s, a))^2\right]. \quad (2)$$
6:  $\quad$ Update $\theta^{t+1} \leftarrow \theta^t + \eta w^t$.
7: **end for**
8: **Return** policy $\widehat{\pi} \sim \text{Uniform}(\{\pi_{\theta^1}, \ldots, \pi_{\theta^{T+1}}\})$.

---

## 5 THEORETICAL ANALYSIS

In this section, we first present our main theoretical guarantees for our Hybrid Actor-Critic algorithm (HAC), and then proceed to its variant HNPG that works for parameterized policy classes. We start by stating the main assumptions and definitions for function approximation, the underlying MDP, the offline data distribution $\nu$, and their relation to the prior works. We remark that all our assumptions and definitions are standard, and are frequently used in the RL theory literature (Agarwal et al., 2019).

**Assumption 1** (Realizability). *For any $\pi$, there exists a $f \in \mathcal{F}$ s.t. $\mathbb{E}_\pi[(f(s, a) - Q^\pi(s, a))^2] = 0$.*

We consider the following notion of inherent Bellman Error, that will appear in our bounds.

**Definition 1** (Point-wise Inherent Bellman Error). *We say that $\mathcal{F}$ has a point-wise inherent Bellman error $\varepsilon_{\text{be}}$, if for all $f \in \mathcal{F}$ and policy $\pi$, there exists a $f' \in \mathcal{F}$ such that $\|f' - \mathcal{T}^\pi f\|_\infty \leq \varepsilon_{\text{be}}$.*

Note that when $\varepsilon_{\text{be}} = 0$, the above definition implies that for any $f \in \mathcal{F}$, its Bellman backup $\mathcal{T} f$ is in the class $\mathcal{F}$, i.e. $\mathcal{F}$ is Bellman complete. While, Bellman completeness is a commonly used assumption in both online (Jin et al., 2021; Xie et al., 2022) and offline RL (Munos & Szepesvári, 2008), our results do not require $\varepsilon_{\text{be}} = 0$. In fact, our algorithm enjoys meaningful guarantees, as presented below, even when Bellman completeness does not hold, i.e., $\varepsilon_{\text{be}}$ could be arbitrarily large.

We next define the coverage for the comparator policy $\pi^e$, which is a common tool in the analysis of policy gradient methods (Kakade & Langford, 2002; Agarwal et al., 2021).

**Definition 2** (NPG Coverage). *Given some comparator policy $\pi^e$, we say that it has coverage $C_{\mathrm{npg},\pi^e}$ over $\pi^e$ if for any policy $\pi$, we have $\left\|\frac{d^{\pi^e}}{d^\pi}\right\|_\infty \le C_{\mathrm{npg},\pi^e}$, where $d^{\pi^e}$ is the occupancy measure of $\pi^e$.*

Note that $C_{\mathrm{npg},\pi^e} < \infty$ if the reset distribution $\mu_0$ satisfies $\|d^{\pi^e}/\mu_0\|_\infty < \infty$, which is a standard assumption used in policy optimization literature such as CPI and NPG (Kakade & Langford, 2002; Agarwal et al., 2021). This condition intuitively says that the reset distribution has good coverage over $d^{\pi^e}$, making it possible to transfer the square error under $d^\pi$ of any policy $\pi$ to $d^{\pi^e}$ (since we always have that $\|\mu_0/d^\pi\|_\infty < 1/1-\gamma$ for all $\pi$, by definition of $\mu_0$). Finally, we introduce the Bellman error transfer coefficient, which allows us to control the expected Belmman under a policy $\pi$ in terms of the squared Bellman error under the offline distribution $\nu$.

**Definition 3** (Bellman error transfer coefficient). *Given the offline distribution $\nu$, for any policy $\pi^e$, we define the Bellman error transfer coefficient as*

$$C_{\mathrm{off},\pi^e} := \max\left\{0, \ \max_\pi \max_{f \in \mathcal{F}} \frac{\mathbb{E}_{s,a\sim d^{\pi^e}}\left[\mathcal{T}^\pi f(s,a) - f(s,a)\right]}{\sqrt{\mathbb{E}_{s,a\sim\nu}\left(\mathcal{T}^\pi f(s,a) - f(s,a)\right)^2}}\right\},$$

*where the $\max_\pi$ is taken over the set of all stationary policies.*

The Bellman error transfer coefficient above was introduced in Song et al. (2023), and is known to be weaker than other related notions considered in prior works, including density ratio (Kakade & Langford, 2002; Munos & Szepesvári, 2008; Chen & Jiang, 2019; Uehara & Sun, 2021), all-policy concentrability coefficient (Munos & Szepesvári, 2008; Chen & Jiang, 2019), square Bellman error based concentrability coefficient (Xie et al., 2021), relative condition number for linear MDP (Uehara et al., 2021; Zhang et al., 2022a), etc. (see Song et al. (2023) for a detailed comparison). Our definition of Bellman error transfer coefficient involves two policies $\pi^e$ and $\pi$, where $\pi^e$ denotes the comparator policy that we wish to compete with (and is thus fixed), and $\pi$ is used to define the Bellman backups (i.e. the terms $\mathcal{T}^\pi f_{h+1}(s,a) - f_h(s,a)$) that we transfer from the offline distribution $\nu$ to the occupancy measure induced by $\pi^e$. We take a $\max$ w.r.t. all possible $\pi$ for the underlying MDP as our analysis proceeds by transferring (from under $\nu$ to $d^{\pi^e}$) the Bellman error terms corresponding to the policies that are generated by our algorithm, which could be arbitrary.

**Theorem 1** (Cumulative suboptimality). *Fix any $\delta \in (0,1)$, and let $\nu$ be an offline data distribution. Suppose the function class $\mathcal{F}$ satisfies Assumption 1. Additionally, suppose that the subroutine HPE is run with parameters $K_1 = 4\lceil\log(1/\gamma)\rceil$, $K_2 = K_1 + T$, and $m_{\mathrm{off}} = m_{\mathrm{on}} = \frac{2T\log(2|\mathcal{F}|/\delta)}{(1-\gamma)^2}$. Then, with probability at least $1 - \delta$, HAC satisfies the following bounds on cumulative suboptimality w.r.t. any comparator policy $\pi^e$:*

- *Under approximate Bellman Complete (when $\varepsilon_{\mathrm{be}} \le 1/T$):*

$$\sum_{t=1}^T V^{\pi^e} - V^{\pi^t} \le \mathcal{O}\left(\frac{1}{(1-\gamma)^2}\sqrt{\log(A)T} + \frac{1}{(1-\gamma)^2}\sqrt{\min\left\{C_{\mathrm{npg},\pi^e}, C_{\mathrm{off},\pi^e}^2\right\}\cdot T}\right).$$

- *Without Bellman Completeness (when $\varepsilon_{\mathrm{be}} > 1/T$):*

$$\sum_{t=1}^T V^{\pi^e} - V^{\pi^t} \le \mathcal{O}\left(\frac{1}{(1-\gamma)^2}\sqrt{\log(A)T} + \frac{1}{(1-\gamma)^2}\sqrt{C_{\mathrm{npg},\pi^e}T}\right).$$

*where $\pi^t$ denotes the policy at round $t$.*

The above shows that as $T$ increases, the average cumulative suboptimality $(\sum_{t=1}^T V^{\pi^e} - V^{\pi^t})/T$ converges to 0 at rate at least $\mathcal{O}(1/\sqrt{T})$. Thus, our algorithm will eventually learn to compete with any comparator policy $\pi^e$ that has bounded $C_{\mathrm{npg},\pi^e}$ (or bounded $C_{\mathrm{off},\pi^e}$ with $\varepsilon_{\mathrm{be}} \le 1/T$). Furthermore, our algorithm exhibits a best-of-both-worlds behavior in the sense that it can operate both with or without approximate Bellman Completeness and enjoys a meaningful guarantee in both cases.

In scenarios when approximate Bellman Completeness holds (i.e. $\varepsilon_{\mathrm{be}} \le 1/T$), the above theorem shows that our algorithm can benefit from access to offline data, and can compete with any comparator policy $\pi^e$ that has a small Bellman error transfer coefficient. This style of bound is typically obtained in pure offline RL by using pessimism, which is typically computationally inefficient (Uehara & Sun, 2021; Xie et al., 2021). In comparison, our algorithm only relies on simple primitives like square-loss regression, which can be made computationally efficient under mild assumptions on $\mathcal{F}$ (see discussion below); On the practical side, least square regression is much easier to implement

and is even compatible with modern neural networks. Finally, note that, under approximate Bellman Completeness and when $C_{\mathrm{off},\pi^e}^2 \leq C_{\mathrm{npg},\pi^e}$, while our guarantees are similar to that of HyQ algorithm from Song et al. (2023), the performance guarantee for HyQ only holds under the conditions that Bellman completeness (when $\varepsilon_{\mathrm{be}}$ is small) and the problem has a small bilinear rank (Du et al., 2021). In comparison, our algorithm enjoys an on-policy NPG style convergence guarantee even when $\varepsilon_{\mathrm{be}}$ or bilinear-rank is large.

When there is no control on the inherent Bellman error, the second bound above holds. Such a bound is typical for policy gradient style algorithms, which do not require any control on $\varepsilon_{\mathrm{be}}$. Again our result is doubly robust in the sense that we still obtain meaningful guarantees when the offline condition does not hold, while previous hybrid RL results like Song et al. (2023) do not have any guarantee when the offline assumptions (that $C_{\mathrm{off},\pi^e}$ is small for some reasonable $\pi^e$ or $\varepsilon_{\mathrm{be}}$ is small) are not met.

Setting $\pi^e$ to be $\pi^\star$ (the optimal policy for the MDP), and using a standard online-to-batch conversion, the above bound implies a sample complexity guarantee for Algorithm 1 for finding an $\varepsilon$-suboptimal policy w.r.t. $\pi^\star$. Details are deferred to Section C.1.5.

On the computation side, there are two key steps that need careful consideration: (a) First, the sampling step in line 5 in HAC. Note that for any given $s$, we have that $\pi^t(a \mid s) \propto \exp(\eta \sum_{\tau=1}^{t-1} f^\tau(s,a))$, so for an efficient implementation, we need the ability to efficiently sample from this distribution. When $|\mathcal{A}|$ is small, this can be trivially done via enumeration. However, when $|\mathcal{A}|$ is large, we may need to resort to parameterized policies in HNPG to avoid computing the partition function. (b) Second, the minimization of (1) in HPE to compute $f_k$ given $f_{k-1}$. Note that (1) is a square loss regression problem in $f$, which can be implemented efficiently in practice. In fact, for various function classes $\mathcal{F}$, explicit guarantees for suboptimality/regret for minimizing the square loss in (1) are well known (Rakhlin & Sridharan, 2014). The above demonstrates the benefit of hybrid RL over online RL and offline RL: by leveraging both offline and online data, we can avoid explicit exploration or conservativeness, making algorithms much more computationally tractable.

**Hybrid NPG with Parameterized Policies.** We can provide a similar bound as in Theorem 1 for the HNPG algorithm, given in Algorithm 3, that works wth parameterized policy class $\Pi = \{\pi_\theta \mid \theta \in \Theta\}$. In particular, we show that HNPG exhibits a best-of-best-worlds behavior in the sense that it can operate with/without approximate Bellman Completeness, and in both cases enjoy a meaningful cumulative suboptimality bound. For the theoretical analysis of HNPG, we make additional assumptions that the policy class is well-parameterized, $\log \pi_\theta(a \mid s)$ is smooth w.r.t. the parameter $\theta$, and that $\mathcal{W}$ realizes the appropriate linear critics for all $\pi \in \Pi$; All of these assumptions are standard in the literature on theoretical analysis of NPG algorithms. We defer the exact details of the assumptions, and the cumulative suboptimality bound for HNPG, to Appendix D.

## 6 EXPERIMENTS

In this section, we describe our empirical comparison of HNPG with other state-of-the-art hybrid RL methods on two challenging rich-observation exploration benchmarks with continuous action space. Our experiments are designed to answer the following questions: **(1)** Is HNPG able to leverage offline data to solve hard exploration problems which cannot be easily solved by pure online on-policy PG methods? **(2)** For setting where Bellman Completeness condition does not necessarily hold, is HNPG able to outperform other hybrid RL baselines which only rely on off-policy learning?

**Implementation.** The implementation of HNPG largely follows from Algorithm 3 and the practical implementation recommendations from TRPO (Schulman et al., 2015). We use a two-layer multi-layer perceptron for Q-functions and policies, plus an additional feature extractor for imaged-based environment. Generalized Advantage Estimation (GAE) (Schulman et al., 2018) is used while calculating online advantages. For NPG-based policy updates, we use conjugate gradient algorithm followed by a line search to find the policy update direction. Following standard combination lock algorithms (Song et al., 2023; Zhang et al., 2022b), instead of a discounted setting policy evaluation, we adapt to the finite horizon setting and train separate Q-functions and policies for each timestep. The pseudocode and hyperparameters are provided in Appendix E.

**Baselines.** We compare HNPG with both pure on-policy and hybrid off-policy actor-critic methods. For pure on-policy method, we use TRPO (Schulman et al., 2015) as the baseline. For hybrid off-policy method, we consider RLPD (Ball et al., 2023), a state-of-the-art algorithm in Mujoco benchmarks, and tuned the hyperparameters specifically for this environment (see Appendix E). We tried

Figure 1: Illustration for continuous combination lock and image-based continuous combination lock.

training separate actors and critics for each time step and also training a single large actor and critic shared for all time steps, for both TRPO and RLPD, and report the best variant. We found that using a single actor and critic for RLPD resulted in better performance while the opposite holds for TRPO. Note that imitation learning such as Behavior Cloning (BC) (Bain & Sammut, 1995) and pure offline learning such as Conservative Q-Learning (CQL) (Kumar et al., 2020) have previously been shown to fail on this benchmark (Song et al., 2023). Hybrid Q-learning methods (Hester et al., 2018; Song et al., 2023) or provable online learning methods for block MDP (Du et al., 2019; Misra et al., 2020; Zhang et al., 2022b; Mhammedi et al., 2023) do not apply here due to the continuous action space.

**Offline distribution.** Following Song et al. (2023), we use a suboptimal offline distribution generated by an $\varepsilon$-greedy policy with $1 - \varepsilon$ probability of taking the good action and $\varepsilon$ probability of taking a random action. $\varepsilon$ is taken to be $1/H$ so that this offline distribution has a bounded density ratio for the optimal policy. The size of the offline dataset is set to $50000$, and around 32%-36% of the trajectories get optimal rewards, for $H$ ranging from 5 to 50.

## 6.1 CONTINUOUS COMBLOCK

The left part of Figure 1 provides an illustration of a rich observation continuous Comblock of horizon $H$ (Misra et al., 2020; Zhang et al., 2022b). For each latent state, there is only one good latent action (out of 10 latent actions) that can lead the agent to the good states (green) in the next time step, while taking any of the other 9 actions will lead the agent to a dead state (orange) from which the agent will never be able to move back to good states; The reward is available at the good states in the last time step. Every timestep, the agent does not have direct access to the latent state, instead, it has access to a high-dimensional observation omitted from the latent state. More details can be found in Appendix E.1. This environment is extremely challenging due to the exploration difficulty and also the need to decode latent states from observations, and many popular deep RL baselines are known to fail (Misra et al., 2020). Built on this environment, we further make the action space continuous. We consider a 10-dimensional action space where $a \in \mathbb{R}^{10}$. At each timestep, when the agent chooses a 10-dimensional action $a$, the action is passed through a softmax layer, i.e., $p \propto \exp(a)$ where the distribution $p$ encodes the probability of choosing the 10 latent actions. A latent action is then sampled based on $p$ and the agent transits to the next time step. This continuous Comblock preserves the exploration difficulty where a uniform exploration strategy only has $\exp(-H)$ probability of getting the optimal reward. The continuous action space makes this environment even harder and rules out many baselines that are based on Q-learning scheme (e.g., HyQ from Song et al. (2023))

The sample complexity of our algorithm vs. the baselines are shown in Figure 2; The loss curves are deferred to Appendix E.2. To begin with, we observe that HNPG can reliably solve continuous Comblock up to horizon 50 with mild sample complexity (50k sub-optimal offline samples and around 30m online samples) despite the challenges of continuous action space. In comparison, TRPO is not able to solve even horizon 5 due to the exploration difficulty in the environment. Although RLPD has the benefit of improved sample complexity (detailed in Appendix E) by reusing past online interactions, it can only solve up to horizon 15. To investigate why off-policy methods cannot solve continuous Comblock as reliably as HNPG, we examine the critic loss of HNPG and RLPD for both online and offline samples. Notably, although both methods maintain a relatively stable critic loss on the offline samples, the online critic loss is more volatile for RLPD since it optimizes the TD error (which requires bootstrap from target network) while HNPG optimizes the policy evaluation error (which is a pure supervised learning problem) for online samples. We believe this unstable online critic loss is why off-policy methods fail to learn reliably in this environment.

## 6.2 IMAGE BASED CONTINUOUS COMBLOCKS

To examine the robustness of HNPG when bellman completeness does not necessarily hold, we carry out experiments on a real-world-image-based continuous Comblock, as depicted in the right part of Figure 1. The only difference between an image-based continuous Comblock and a continuous

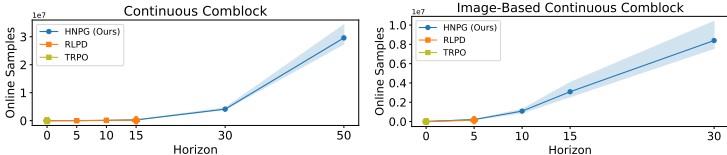

Figure 2: Comparison of sample complexity of different algorithms in Continuous Comblock and Image-Based Continuous Comblock benchmarks. The number of online samples is averaged over 5 random seeds and the standard deviation is shaded.

Comblock lies in their observation spaces. Specifically, for an image-based continuous Comblock, each latent state is represented by a class in cifar100 (Krizhevsky, 2009) and an observation is generated by randomly sampling a training image from that class. After sampling an image, we get the observation by using the "ViT-B/32" CLIP (Radford et al., 2021) image encoder to calculate a pre-trained feature. In addition, in the training environment, the image observations are sampled from the training set of cifar100 while in the test environment the image observations are drawn from the val set of cifar100. Unlike mujoco-based benchmarks where transition is often deterministic and initial state distribution is narrow, our setting, which uses real-world supervised learning datasets with a clear training and testing data split, challenges the algorithms to generalize to unseen test examples.

To get a sense of the inherent bellman error in Definition 1 for this setting, we conduct supervised learning experiments on cifar100 with the same functions as used for actors and critics (on top of the CLIP feature). The resulting top-1 classification accuracy is 77.7% on the training set and 72.1% on the test set, showing that the latent states are not 100% decodable from the pre-trained features using our function class. The fact that our function classes are not rich enough to exactly decode the latent states introduces model misspecification such that Bellman completeness may not hold.

The sample complexity results are shown in Figure 2 (right) and the loss curve results of horizon 5 are shown in Figure 3. First, TRPO fails to solve horizon 5 again due to its inefficient exploration. In this more realistic image-based setting, we observe that RLPD struggles even in horizon 5 and completely fails for horizon 10. In contrast, HNPG not only has a reduced sample complexity for horizon 5 but also reliably solves up to horizon 30 with around 10m online samples. To investigate this contrast, we examine the critic loss of HNPG and RLPD for horizon 5. While the offline critic TD loss stays stable for both HNPG and RLPD, the online critic TD loss is exploding for RLPD. This is not surprising since for environments where the Bellman completeness condition does not hold, Bellman backup based methods can diverge and become unstable to train. On the other hand, the on-policy training loss for HNPG is small since the on-policy training is based on supervised learning style least square regression instead of TD-style bootstrapping.

Finally, the train and test learning curves are also reported in Figure 3. It is observed that both the training curve and the test curve of HNPG have smaller variances, indicating that it is more stable to train, while those of RLPD have a larger variance, indicating that it is less stable. More importantly, even though two methods reach a similar train set reward in the best random seed, HNPG achieves a larger margin over RLPD in the test environment (around 0.8 compared to 0.6), showing that HNPG is better at generalization since RLPD uses the off-policy algorithm SAC which typically has a much higher updates-to-data (gradient updates per $(s, a, r, s')$ collection) ratio from 1:1 to 10:1, making it possible to overfit to the training data in the replay buffer.

## 7 CONCLUSION

We propose a new actor-critic style algorithm for the hybrid RL setting. Unlike previous model-free hybrid RL methods that only rely on off-policy learning, our proposed algorithms HAC and (the parametrized version) HNPG perform on-policy learning over the online data together with an off-policy learning procedure using the offline data. Thus, our algorithms are able to achieve guarantees that are the best-of-both-worlds. In particular, our algorithms achieve the state-of-art theoretical guarantees of offline RL when offline RL-specific assumptions (e.g., Bellman completeness and offline distribution coverage) hold, while at the same time enjoy the theoretical guarantees of on-policy policy gradient methods regardless of the offline RL assumptions' validity. Our experiment results show that HNPG can indeed outperform the pure on-policy method, and stay robust to the lack of Bellman completeness condition in practice; In the latter scenario, other off-policy hybrid RL algorithms fail. Future research directions include sharpening the rates in our theoretical bounds and trying our algorithmic ideas for large-scale applications.

ACKNOWLEDGEMENTS

We thank Akshay Krishnamurthy and Drew Bagnell for useful discussions. AS acknowledges support from the Simons Foundation and NSF through award DMS-2031883, as well as from the DOE through award DE- SC0022199.

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

CONTENTS OF APPENDIX

**Additional notation.** Given a dataset $\mathcal{D} = \{x\}$, we denote $\widehat{\mathbb{E}}_{\mathcal{D}}[f(x)]$ as its empirical average, i.e., $\widehat{\mathbb{E}}_{\mathcal{D}}[f(x)] = \frac{1}{|\mathcal{D}|} \sum_{x \in \mathcal{D}} [f(x)]$. For any function $f$, and data distribution $\mu$, we define $\|f\|_{2,\mu}^2 = \mathbb{E}_{s,a \sim \mu}[f(s,a)^2]$. Unless explicitly specified, all $\log$ are natural logarithms.

## A  PRELIMINARIES

The following is the well-known Performance Difference Lemma.

**Lemma 1** (Performance difference lemma; Kakade & Langford (2002))**.** *For any two policies $\pi$, $\pi'$,*

$$V^\pi - V^{\pi'} = \frac{1}{1-\gamma} \mathbb{E}_{s,a \sim d^\pi} \left[ A^{\pi'}(s,a) \right],$$

*where $V^\pi = \mathbb{E}_{s,a \sim \mu_0}[Q^\pi(s,a)]$, and $A^{\pi'}(s,a) = Q^{\pi'}(s,a) - V^{\pi'}(s)$.*

**Lemma 2.** *For any policy $\pi$, and non-negative function $g(s,a)$, we have:*

(a)  $\mathbb{E}_{s,a \sim \mu_0}[g(s,a)] \leq \frac{1}{1-\gamma} \mathbb{E}_{s,a \sim d^\pi}[g(s,a)]$.

(b)  $\mathbb{E}_{\bar{s},\bar{a} \sim d^\pi} \mathbb{E}_{s \sim P(\cdot|\bar{s},\bar{a}), a \sim \pi(a|s)}[g(s,a)] \leq \frac{1}{\gamma} \mathbb{E}_{s,a \sim d^\pi}[g(s,a)]$.

*where $\mu_0$ denotes the initial reset distribution (which is the same for all policies $\pi$).*

*Proof.* We prove the two parts separately:

(a)  The proof follows trivially from the definition of

$$d^\pi(s,a) = (1-\gamma)\left( \mu_0(s,a) + \sum_{h=1}^{\infty} \gamma^h d_h^\pi(s,a) \right).$$

(b)  Recall that $\lim_{h \to \infty} \gamma^h = 0$. We start by noting that:

$$d^\pi(s,a) = (1-\gamma)(\mu_0(s,a) + \gamma d_1^\pi(s,a) + \gamma^2 d_2^\pi(s,a) + \dots) \tag{3}$$

$$\geq \gamma(1-\gamma)\left( \sum_{\bar{s},\bar{a}} \mu_0(\bar{s},\bar{a})P(s|\bar{s},\bar{a})\pi(a|s) + \gamma \sum_{\bar{s},\bar{a}} d_1^\pi(\bar{s},\bar{a})P(s|\bar{s},\bar{a})\pi(a|s) + \dots \right)$$

$$= \gamma(1-\gamma) \sum_{\bar{s},\bar{a}} \left( \mu_0(\bar{s},\bar{a}) + \gamma d_1^\pi(\bar{s},\bar{a}) + \dots \right) P(s|\bar{s},\bar{a})\pi(a|s)$$

$$= \gamma \sum_{\bar{s},\bar{a}} d^\pi(\bar{s},\bar{a})P(s|\bar{s},\bar{a})\pi(a|s) \tag{4}$$

$$= \gamma \mathbb{E}_{\bar{s},\bar{a} \sim d^\pi}[P(s|\bar{s},\bar{a})\pi(a|s)],$$

where (4) follows by plugging in the relation (3) for $\bar{s}, \bar{a}$. The above implies that for any function $g \geq 0$,

$$\sum_{s,a} d^\pi(s,a)g(s,a) \geq \sum_{s,a} \gamma \mathbb{E}_{\bar{s},\bar{a} \sim d^\pi}[P(s|\bar{s},\bar{a})\pi(a|s)g(s,a)],$$

which implies that

$$\mathbb{E}_{\bar{s},\bar{a} \sim d^\pi} \mathbb{E}_{s \sim P(\cdot|\bar{s},\bar{a}), a \sim \pi(a|s)}[g(s,a)] \leq \frac{1}{\gamma} \mathbb{E}_{s,a \sim d^\pi}[g(s,a)].$$

$\square$

The next lemma bounds the gap between the value of the policy $\pi^e$ and policy $\pi$ (given a value function $f$) in terms of the expected bellman error of $f$ and the gap between $f(s, \pi(s))$ and $f(s, \pi^e(s))$. [3]

---

[3]In order to keep the notation simple, for stochastic policies $\pi$, we define $f(s, \pi(s)) = \mathbb{E}_{a \sim \pi(s)}[f(s,a)]$.

**Lemma 3.** *For any policy $\pi$, value function $f$, and comparator policy $\pi^e$,*

$$V^{\pi^e} - V^\pi \leq \mathbb{E}_{s_0,a_0 \sim \mu_0}[|f(s_0, \pi(s_0)) - Q^\pi(s_0, a_0)|]$$

$$+ \frac{1}{1-\gamma} \mathbb{E}_{s,a \sim d^{\pi^e}}[(\mathcal{T}^\pi f(s,a) - f(s,a))] + \frac{1}{1-\gamma} \mathbb{E}_{s,a \sim d^{\pi^e}}[f(s,a) - f(s, \pi(s))],$$

*where for any policy $\pi$ we define $V^\pi = \mathbb{E}_{s,a \sim d^\pi}[Q^\pi(s,a)]$.*

*Proof.* We start by noting that

$$V^{\pi^e} = \mathbb{E}_{s_0,a_0 \sim \mu_0}\left[Q^{\pi^e}(s_0, a_0)\right]$$

$$= \mathbb{E}_{s_0,a_0 \sim \mu_0}[r(s_0, \pi^e(s_0))] + \mathbb{E}_{s_0,a_0 \sim \mu_0, s_1 \sim P(\cdot|s_0,a_0)}\left[V^{\pi^e}(s_1)\right]$$

$$= \mathbb{E}_{s_0,a_0 \sim \mu_0}\left[r(s_0, a_0) + \gamma \mathbb{E}_{s_1 \sim P(\cdot|s_0,a_0)}[V^{\pi^e}(s_1)]\right.$$
$$\left. - \mathcal{T}^\pi f(s_0, a_0) + \mathcal{T}^\pi f(s_0, a_0)\right]$$

$$= \gamma \mathbb{E}_{s_1 \sim d_1^{\pi^e}}\left[V^{\pi^e}(s_1) - f(s_1, \pi(s_1))\right] + \mathbb{E}_{s_0,a_0 \sim \mu_0}[\mathcal{T}^\pi f(s_0, a_0)],$$

where the last line follows from the definition of $T^\pi f$, and the fact that $s_0, a_0 \sim \mu_0$ followed by $s_1 \sim P(\cdot \mid s_0, a_0)$ is equivalent to $s_1 \sim d_0^{\pi^e}$, by definition. Using the fact that $\mu_0 = d_0^{\pi^e}$, and the definition of $d_1^{\pi^e}$, we get

$$\mathbb{E}_{s_0,a_0 \sim \mu_0}\left[Q^{\pi^e}(s_0, a_0) - f(s_0, \pi(s_0))\right] = \mathbb{E}_{s_0,a_0 \sim d_0^{\pi^e}}\left[Q^{\pi^e}(s_0, a_0) - f(s_0, \pi(s_0))\right]$$

$$= \gamma \mathbb{E}_{s_1,a_1 \sim d_1^{\pi^e}}\left[Q^{\pi^e}(s_1, a_1) - f(s_1, \pi(s_1))\right]$$
$$+ \mathbb{E}_{s_0,a_0 \sim d_0^{\pi^e}}[\mathcal{T}^\pi f(s_0, a_0) - f(s_0, a_0)]$$
$$+ \mathbb{E}_{s_0,a_0 \sim d_0^{\pi^e}}[f(s_0, a_0) - f(s_0, \pi(a_0))]$$

Repeating the above expansion for the first term in the above, and then recursively for all future terms, along with the fact that $\lim_{h \to \infty} \gamma^h = 0$, we get that

$$V^{\pi^e} - \mathbb{E}_{s_0 \sim \mu_0}[f(s_0, \pi(s_0))] \leq \sum_{h=0}^{\infty} \gamma^h \mathbb{E}_{s,a \sim d_h^{\pi^e}}[(\mathcal{T}^\pi f(s,a) - f(s,a))]$$

$$+ \sum_{h=0}^{\infty} \gamma^h \mathbb{E}_{s,a \sim d_h^{\pi^e}}[f(s,a) - f(s, \pi(s))].$$

The above implies that

$$V^{\pi^e} - V^\pi = V^{\pi^e} - \mathbb{E}_{s_0 \sim \mu_0}[f(s_0, \pi(s_0))] + \mathbb{E}_{s_0,a_0 \sim \mu_0}[f(s_0, \pi(s_0)) - Q^\pi(s_0, a_0)]$$
$$\leq \mathbb{E}_{s_0,a_0 \sim \mu_0}[|f(s_0, \pi(s_0)) - Q^\pi(s_0, a_0)|]$$

$$+ \sum_{h=0}^{\infty} \gamma^h \mathbb{E}_{s,a \sim d_h^{\pi^e}}[(\mathcal{T}^\pi f(s,a) - f(s,a))]$$

$$+ \sum_{h=0}^{\infty} \gamma^h \mathbb{E}_{s,a \sim d_h^{\pi^e}}[f(s,a) - f(s, \pi(s))].$$

$\square$

The following is a well-known generalization bound that holds for least squares regression. We recall the version given in Song et al. (2023), and skip the proof for conciseness.

**Lemma 4** (Least squares generalization bound, (Song et al., 2023, Lemma 3)). *Let $R > 0$, $\delta \in (0,1)$, and consider a sequential function estimation setting with an instance space $\mathcal{X}$ and target space $\mathcal{Y}$. Let $\mathcal{H} : \mathcal{X} \mapsto [-R, R]$ be a class of real valued functions. Let $\mathcal{D} = \{(x_1, y_1), \ldots, (x_T, y_T)\}$ be a dataset of $T$ points where $x_t \sim \rho_t := \rho_t(x_{1:t-1}, y_{1:t-1})$, and $y_t$ is sampled via the conditional probability $p_t(x_t)$ (which could be adversarially chosen). Additionally, suppose that $\max_t |y_t| \leq R$ and $\max_h \max_x |h(x)| \leq R$. Then, the least square solution $\widehat{h} \leftarrow \arg\min_{h \in \mathcal{H}} \sum_{t=1}^{T}(h(x_t) - y_t)^2$*

*satisfies*

$$\sum_{t=1}^{T} \mathbb{E}_{x\sim\rho_t, y\sim p_t(x)}\left[(\widehat{h}(x) - y)^2\right] \leq \inf_{h\in\mathcal{H}} \sum_{t=1}^{T} \mathbb{E}_{x\sim\rho_t, y\sim p_t(x)}\left[(h(x) - y)^2\right] + 256R^2 \log(2|\mathcal{H}|/\delta)$$

*with probability at least $1 - \delta$.*

## B  HYBRID FITTED POLICY EVALUATION ANALYSIS

In this section, we provide our main technical result for the HPE subroutine in Algorithm 2. In particular, we show that for any input policy $\pi$, HPE succeeds in finding a value function $\bar{f}$ that closely approximates $Q^\pi$ on the state action distribution $d^\pi$, and at the same time has small Bellman error (w.r.t. backups $\mathcal{T}^\pi$) on the offline data distribution $\nu$. The former guarantee is used for on-policy online analysis, and the latter part is used for the Hybrid analysis.

**Lemma 5.** *Suppose that* HPE *procedure, given in Algorithm 2, is executed for a policy $\pi$ with parameters $K_1 = 1 + \left\lceil \log_\gamma\left(\frac{6\lambda\varepsilon_{\mathrm{be}} + \varepsilon_{\mathrm{stat}}(1-\gamma)}{\lambda^2}\right)\right\rceil$, $K_2 = K_1 + \left\lceil \frac{1}{20\varepsilon_{\mathrm{stat}}(1-\gamma)}\right\rceil$ and $m_{\mathrm{off}} = m_{\mathrm{on}} = \frac{2T\log(|2\mathcal{F}|/\delta)}{(1-\gamma)^2}$. Then, with probability at least $1 - \delta$, the output value function $\bar{f}$ satisfies*

*(a)* $\mathbb{E}_{s,a\sim d^\pi}[(\bar{f}(s,a) - Q^\pi(s,a))^2] \leq \frac{12}{(1-\gamma)^2} \min\{\frac{1}{\lambda}, \varepsilon_{\mathrm{be}}\} + \frac{12\varepsilon_{\mathrm{stat}}}{\lambda(1-\gamma)} =: \Delta_{\mathrm{on}},$

*(b)* $\mathbb{E}_{s,a\sim\nu}[(\bar{f}(s,a) - \mathcal{T}^\pi\bar{f}(s,a))^2] \leq \frac{40}{(1-\gamma)}\left(\frac{\lambda\varepsilon_{\mathrm{be}}}{1-\gamma} + \varepsilon_{\mathrm{stat}}\right) =: \Delta_{\mathrm{off}},$

*where $\varepsilon_{\mathrm{stat}} = 128(1+\lambda)/T$.*

*Proof of Lemma 5.* We first define additional notation. For the $k$-th iteration in Algorithm 2, let

$$\widehat{L}_k(f) := \widehat{\mathbb{E}}_{s,a,s'\sim\mathcal{D}_k^\nu, a'\sim\pi(\cdot|s')}[(f(s,a) - r - \gamma f_{k-1}(s',a'))^2] + \lambda\widehat{\mathbb{E}}_{s,a,y\sim\mathcal{D}_k^\pi}[(f(s,a) - y)^2], \tag{5}$$

and,

$$L_k(f) := \mathbb{E}_{s,a,s'\sim\nu, a'\sim\pi(\cdot|s')}[(f(s,a) - r - \gamma f_{k-1}(s',a'))^2] + \lambda\mathbb{E}_{s,a,y\sim d^\pi}[(f(s,a) - y)^2].$$

Thus, an application of Lemma 4 implies that the optimization procedure in (1) satisfies, with probability at least $1 - \delta$, the guarantee

$$L(f_k) \leq \min_{f\in\mathcal{F}} L(f) + \varepsilon_{\mathrm{stat}}, \tag{6}$$

where

$$\varepsilon_{\mathrm{stat}} \leq 256 \cdot \frac{1+\lambda}{(1-\gamma)^2} \cdot \frac{\ln(2|\mathcal{F}|/\delta)}{\min\{m_{\mathrm{on}}, m_{\mathrm{off}}\}} \leq 256 \cdot \frac{1+\lambda}{2T}, \tag{7}$$

since the terms in the least squares optimization problem given by the objective in (5) satisfies the bound $R^2 \leq (1+\lambda)\sup_{s,a}|f(s,a)|^2 \leq (1+\lambda)/(1-\gamma^2)$.

Now, fix any $k \in [K_1 + K_2]$, and define a function $\widetilde{f}_k \in \mathcal{F}$ such that

$$\|\widetilde{f}_k - \mathcal{T}^\pi f_{k-1}\|_\infty \leq \varepsilon_{\mathrm{be}}, \tag{8}$$

which is guaranteed to exist from Definition 1. Plugging in $\widetilde{f}_k$ instead of the corresponding minimizer in the RHS of (6), we get that

$$\mathbb{E}_{s,a,s'\sim\nu, a'\sim\pi(\cdot|s')}[(f_k(s,a) - r - \gamma f_{k-1}(s',a'))^2] + \lambda\mathbb{E}_{s,a,y\sim d^\pi}[(f_k(s,a) - y)^2]$$

$$\leq \mathbb{E}_{s,a,s'\sim\nu, a'\sim\pi(\cdot|s')}[(\widetilde{f}_k(s,a) - r - \gamma f_{k-1}(s',a'))^2] + \lambda\mathbb{E}_{s,a,y\sim d^\pi}[(\widetilde{f}_k(s,a) - y)^2] + \varepsilon_{\mathrm{stat}}.$$

Due to the linearity of expectation, and adding appropriate terms on both the sides to handle the variance, we get that

$$\mathbb{E}_{s,a\sim\nu}[(f_k(s,a) - \mathcal{T}^\pi f_{k-1}(s,a))^2] + \lambda\mathbb{E}_{s,a\sim d^\pi}[(f_k(s,a) - Q^\pi(s,a))^2]$$

$$\leq \mathbb{E}_{s,a\sim\nu}[(\widetilde{f}_k(s,a) - \mathcal{T}^\pi f_{k-1}(s,a))^2] + \lambda\mathbb{E}_{s,a\sim d^\pi}[(\widetilde{f}_k(s,a) - Q^\pi(s,a))^2] + \varepsilon_{\mathrm{stat}} \tag{9}$$

$$\leq \varepsilon_{\mathrm{be}}^2 + \lambda\mathbb{E}_{s,a\sim d^\pi}[(\widetilde{f}_k(s,a) - Q^\pi(s,a))^2] + \varepsilon_{\mathrm{stat}}$$

$$\leq \frac{2\varepsilon_{\text{be}}}{1-\gamma} + \lambda \mathbb{E}_{s,a\sim d^\pi}[(\widetilde{f}_k(s,a) - Q^\pi(s,a))^2] + \varepsilon_{\text{stat}},$$

where the second last line follows from (8) and the last line uses the fact that $\varepsilon_{\text{be}} \leq 2/(1-\gamma)$ since $\max(s,a)|f(s,a) - f'(s,a)| \leq \frac{2}{1-\gamma}$ for any $f$ and $f' \in \mathcal{F}$. The above implies that

$$\mathbb{E}_{s,a\sim\nu}[(f_k(s,a) - \mathcal{T}^\pi f_{k-1}(s,a))^2] \leq \lambda \mathbb{E}_{s,a\sim d^\pi}[(\widetilde{f}_k(s,a) - Q^\pi(s,a))^2] + \frac{2\varepsilon_{\text{be}}}{1-\gamma} + \varepsilon_{\text{stat}}, \quad (10)$$

and

$$\lambda \mathbb{E}_{s,a\sim d^\pi}[(f_k(s,a) - Q^\pi(s,a))^2] \leq \lambda \mathbb{E}_{s,a\sim d^\pi}[(\widetilde{f}_k(s,a) - Q^\pi(s,a))^2] + \frac{2\varepsilon_{\text{be}}}{1-\gamma} + \varepsilon_{\text{stat}}. \quad (11)$$

We next focus our attention on bounding the first term in the RHS above. Note that

$$\mathbb{E}_{s,a\sim d^\pi}[(\widetilde{f}_k(s,a) - Q^\pi(s,a))^2]$$

$$= \mathbb{E}_{s,a\sim d^\pi}\left[(\widetilde{f}_k(s,a) - \mathcal{T}^\pi f_{k-1}(s,a))^2\right.$$

$$\left. + (\mathcal{T}^\pi f_{k-1}(s,a) - Q^\pi(s,a))(2\widetilde{f}_k(s,a) - \mathcal{T}^\pi f_{k-1}(s,a) - Q^\pi(s,a))\right]$$

$$\overset{(i)}{\leq} \frac{2\varepsilon_{\text{be}}}{1-\gamma} + 2\mathbb{E}_{s,a\sim d^\pi}[(\mathcal{T}^\pi f_{k-1}(s,a) - Q^\pi(s,a))(\widetilde{f}_k(s,a) - \mathcal{T}^\pi f_{k-1}(s,a))]$$

$$+ \mathbb{E}_{s,a\sim d^\pi}\left[(\mathcal{T}^\pi f_{k-1}(s,a) - Q^\pi(s,a))^2\right]$$

$$\overset{(ii)}{\leq} \frac{4\varepsilon_{\text{be}}}{1-\gamma} + \mathbb{E}_{s,a\sim d^\pi}\left[(\mathcal{T}^\pi f_{k-1}(s,a) - Q^\pi(s,a))^2\right] \quad (12)$$

where $(i)$ follows from (8), and a simple manipulation of the second term, and $(ii)$ again uses (8) and the fact that $\sup_{f,\pi} \max\{\|f\|_\infty, \|\mathcal{T}^\pi f\|_\infty\} \leq 1/(1-\gamma)$. For the second term above, we have

$$\mathbb{E}_{s,a\sim d^\pi}\left[(\mathcal{T}^\pi f_{k-1}(s,a) - Q^\pi(s,a))^2\right] = \gamma^2 \mathbb{E}_{s,a\sim d^\pi, s'\sim P(\cdot|s,a), a'\sim\pi(s')}\left[(f_{k-1}(s',a') - Q^\pi(s',a'))^2\right]$$

$$\leq \gamma \mathbb{E}_{s,a\sim d^\pi}\left[(f_{k-1}(s,a) - Q^\pi(s,a))^2\right],$$

where the second inequality is due to Lemma 2. Plugging this bound back in (12), we get that

$$\mathbb{E}_{s,a\sim d^\pi}[(\widetilde{f}_k(s,a) - Q^\pi(s,a))^2] \leq \frac{4\varepsilon_{\text{be}}}{1-\gamma} + \gamma \mathbb{E}_{s,a\sim d^\pi}\left[(f_{k-1}(s,a) - Q^\pi(s,a))^2\right]. \quad (13)$$

We now complete the bounds on (10) and (11), using the bound in (13).

- *Bound on* (11): Using the relation (13) in (11), we get

$$\mathbb{E}_{s,a\sim d^\pi}[(f_k(s,a) - Q^\pi(s,a))^2] \leq \gamma \mathbb{E}_{s,a\sim d^\pi}\left[(f_{k-1}(s,a) - Q^\pi(s,a))^2\right] + \frac{6\varepsilon_{\text{be}}}{1-\gamma} + \frac{\varepsilon_{\text{stat}}}{\lambda},$$

  where we simplified the RHS since $\gamma \leq 1$. Recursing the above relation from $k-1$ to 1, we get that

$$\mathbb{E}_{s,a\sim d^\pi}(f_k(s,a) - Q^\pi(s,a))^2 \leq \gamma^{k-1}\mathbb{E}_{s,a\sim d^\pi}\left[(f_1(s,a) - Q^\pi(s,a))^2\right] + \frac{1}{1-\gamma}\left(\frac{6\varepsilon_{\text{be}}}{1-\gamma} + \frac{\varepsilon_{\text{stat}}}{\lambda}\right)$$

$$\leq \frac{\gamma^{k-1}}{(1-\gamma)^2} + \frac{1}{1-\gamma}\left(\frac{6\varepsilon_{\text{be}}}{1-\gamma} + \frac{\varepsilon_{\text{stat}}}{\lambda}\right),$$

  where the last line uses the fact that $\sup_{s,a}|f(s,a)| \leq 1/1-\gamma$. Since, the above inequality holds for all $k \leq K_1 + K_2$, setting $K_1 = \left\lceil \log_\gamma\left(\frac{6\lambda\varepsilon_{\text{be}} + \varepsilon_{\text{stat}}(1-\gamma)}{\lambda}\right)\right\rceil + 1$, we get that

$$\forall k \in [K_1, K_2]: \quad \mathbb{E}_{s,a\sim d^\pi}[(f_k(s,a) - Q^\pi(s,a))^2] \leq \frac{12}{(1-\gamma)}\left(\frac{\varepsilon_{\text{be}}}{1-\gamma} + \frac{\varepsilon_{\text{stat}}}{\lambda}\right). \quad (14)$$

- *Bound on* (10): Using the bound (14) in (13), we get that

$$\forall k \in [K_1 + 1, K_2]: \quad \mathbb{E}_{s,a\sim d^\pi}[(\widetilde{f}_k(s,a) - Q^\pi(s,a))^2] \leq \frac{16}{(1-\gamma)}\left(\frac{\varepsilon_{\text{be}}}{1-\gamma} + \frac{\gamma\varepsilon_{\text{stat}}}{\lambda}\right).$$

  Using the above relation in (10), we get that for all $K_1 + 1 \leq k \leq K_2$,

$$\mathbb{E}_{s,a\sim\nu}[(f_k(s,a) - \mathcal{T}^\pi f_{k-1}(s,a))^2] \leq \frac{20}{(1-\gamma)}\left(\frac{\lambda\varepsilon_{\text{be}}}{1-\gamma} + \varepsilon_{\text{stat}}\right), \quad (15)$$

where again we used the fact that $\gamma \in (0, 1)$.

- *An alternate bound on* (11). We now provide an alternate bound on (11) through an independent analysis. Let $\widetilde{f}_k = Q^\pi$, which is guaranteed to be in the class $\mathcal{F}$ due to Assumption 1. Thus, repeating the same steps till (9) but with this choice of $\widetilde{f}_k$, we get that

$$\mathbb{E}_{s,a\sim\nu}[(f_k(s,a) - \mathcal{T}^\pi f_{k-1}(s',a'))^2] + \lambda\mathbb{E}_{s,a\sim d^\pi}[(f_k(s,a) - Q^\pi(s,a))^2]$$
$$\leq \mathbb{E}_{s,a\sim\nu}(Q^\pi(s,a) - \mathcal{T}^\pi f_{k-1}(s,a))^2 + \varepsilon_{\text{stat}}.$$

Ignoring positive terms in the LHS, the above implies that

$$\mathbb{E}_{s,a\sim d^\pi}[(f_k(s,a) - Q^\pi(s,a))^2] \leq \frac{1}{\lambda}\mathbb{E}_{s,a\sim\nu}[(Q^\pi(s,a) - T^\pi f_{k-1}(s,a))^2] + \frac{\varepsilon_{\text{stat}}}{\lambda}$$
$$\leq \frac{1}{\lambda(1-\gamma)^2} + \frac{\varepsilon_{\text{stat}}}{\lambda}. \tag{16}$$

Combining the above results, we note that for all $K_1 + 1 \leq k \leq K_2$,

$$\mathbb{E}_{s,a\sim d^\pi}[(f_k(s,a) - Q^\pi(s,a))^2] \leq \frac{12}{(1-\gamma)^2}\min\left\{\frac{1}{\lambda}, \varepsilon_{\text{be}}\right\} + \frac{12\varepsilon_{\text{stat}}}{\lambda(1-\gamma)}, \tag{17}$$

and,

$$\mathbb{E}_{s,a\sim\nu}[(f_k(s,a) - \mathcal{T}^\pi f_{k-1}(s,a))^2] \leq \frac{20}{(1-\gamma)}\left(\frac{\lambda\varepsilon_{\text{be}}}{1-\gamma} + \varepsilon_{\text{stat}}\right). \tag{18}$$

We are now ready to complete the proof. Equipped with the bounds in (17) and (18), we note that $\bar{f} = \frac{1}{K_2-K_1}\sum_{k=K_1+1}^{K_2} f_k$ satisfies

$$\mathbb{E}_{s,a\sim d^\pi}[(\bar{f}(s,a) - Q^\pi(s,a))^2] \leq \frac{1}{K_2-K_1}\sum_{k=K_1+1}^{K_2}\mathbb{E}_{s,a\sim d^\pi}[(f_k(s,a) - Q^\pi(s,a))^2]$$
$$\leq \frac{12}{(1-\gamma)^2}\min\left\{\frac{1}{\lambda}, \varepsilon_{\text{be}}\right\} + \frac{12\varepsilon_{\text{stat}}}{\lambda(1-\gamma)}, \tag{19}$$

where the first line follows from Jensen's inequality, and the second line is due to (17). Similarly, we have that

$$\mathbb{E}_{s,a\sim\nu}[(\bar{f}(s,a) - \mathcal{T}^\pi \bar{f}(s,a))^2]$$
$$= \mathbb{E}_{s,a\sim\nu}\left(\frac{1}{K_2-K_1}\left(f_{K_1+1}(s,a) - \mathcal{T}^\pi f_{K_2}(s,a) + \sum_{k=K_1+2}^{K_2} f_k(s,a) - \mathcal{T}^\pi f_{k-1}(s,a)\right)\right)^2$$
$$\leq \frac{1}{K_2-K_1}\mathbb{E}_{s,a\sim\nu}\left((f_{K_1+1}(s,a) - \mathcal{T}^\pi f_{K_2}(s,a))^2 + \sum_{k=K_1+2}^{K_2}(f_k(s,a) - \mathcal{T}^\pi f_{k-1}(s,a))^2\right)$$
$$\leq \frac{1}{K_2-K_1}\mathbb{E}_{s,a\sim\nu}\left(\frac{1}{(1-\gamma)^2} + \sum_{k=K_1+2}^{K_2}(f_k(s,a) - \mathcal{T}^\pi f_{k-1}(s,a))^2\right)$$
$$\leq \frac{1}{(K_2-K_1)(1-\gamma^2)} + \frac{20}{(1-\gamma)}\left(\frac{\lambda\varepsilon_{\text{be}}}{1-\gamma} + \varepsilon_{\text{stat}}\right), \tag{20}$$

where the first inequality is an application of Jensen's inequality, second inequality simply plus in the fact that $\max\|f\|_\infty, \|\mathcal{T}^\pi f\|_\infty \leq 1/(1-\gamma)$, and the last line simply plugs in (18). Setting

$$K_2 = K_1 + \frac{1}{20\varepsilon_{\text{stat}}(1-\gamma)}.$$

completes the proof. $\qquad\square$

## C    MISSING PROOFS FROM SECTION 4

### C.1    PROOF OF THEOREM 1

#### C.1.1    SUPPORTING TECHNICAL RESULTS

We first provide a useful technical result. In the analysis, $f^t(s,a) - f^t(s, \pi^t(s))$ will represent an approximation for the advantage function $A^{\pi^t}(s,a)$. The following bounds the expected advantage when $s, a$ are sampled from the occupancy of $\pi^e$.

**Lemma 6.** *Supppose $f^t$ and $\pi^t$ denote the value function and the policies at round $t$ in Algorithm 1. Then, for any $\eta \leq (1-\gamma)/2$ and policy $\pi^e$, we have*

$$\sum_t \mathbb{E}_{s,a \sim d^{\pi^e}}[f^t(s,a) - f^t(s, \pi^t(s))] \leq \frac{2}{1-\gamma}\sqrt{\log(A)T}.$$

*Proof.* For the ease of notation, let $\bar{f}^t(s,a) := f^t(s,a) - f^t(s, \pi^t(s))$. Recall that the policy $\pi_{t+1}$, after round $t$, is defined as

$$\pi^{t+1}(a \mid s) = \frac{\pi^t(a \mid s)\exp(\eta \bar{f}^t(s,a))}{\sum_{a'}\pi^t(a' \mid s)\exp(\eta \bar{f}^t(s,a'))}.$$

Let $Z_t = \sum_{a'}\pi^t(a' \mid s)\exp(\eta \bar{f}^t(s,a'))$ be the normalization constant, and note that for any $s$,

$$\mathbb{E}_{a \sim \pi^e(s)}\left[\log \pi^t(a \mid s) - \log \pi^{t+1}(a \mid s)\right] = \mathbb{E}_{a \sim \pi^e(s)}\left[-\eta \bar{f}^t(s,a) + \log(Z_t)\right]. \qquad (21)$$

We first bound the term comprising of $\log(Z_t)$. Note that since $\eta \leq (1-\gamma)/2$ and $\|f\|_\infty \leq 1/(1-\gamma)$, we have that $\eta \bar{f}^t(s,a) \leq 1$. Thus, using the fact that $\exp(x) \leq 1 + x + x^2$ for any $x \leq 1$, we have

$$\log(Z_t) = \log\left(\sum_{a'}\pi^t(a' \mid s)\exp(\eta \bar{f}^t(s,a'))\right)$$

$$\leq \log\left(\sum_{a'}\pi^t(a' \mid s)(1 + \eta \bar{f}^t(s,a') + \eta^2 \bar{f}^t(s,a')^2)\right)$$

$$\leq \log\left(1 + \frac{\eta^2}{(1-\gamma)^2}\right) \leq \frac{\eta^2}{(1-\gamma)^2},$$

where in the second last inequality we use the fact that $\sum_{a'}\pi^t(a' \mid s)\bar{f}^t(s,a) = 0$ by the definition of $\bar{f}^t$, and that $\|f\|_\infty \leq 1/(1-\gamma)$. The last inequality simply uses the fact that $\log(1+x) \leq x$ for any $x \geq 0$. Plugging in the above bound in (21), and rearranging the terms, we get that

$$\mathbb{E}_{a \sim \pi^e(s)}\left[\bar{f}^t(s,a)\right] \leq \frac{1}{\eta}\mathbb{E}_{a \sim \pi^e(s)}\left[\log \pi^{t+1}(a \mid s) - \log \pi^t(a \mid s)\right] + \frac{\eta}{(1-\gamma)^2}.$$

Telescoping the above for $t$ from 1 to $T$, and using the fact that $\log(\pi(a \mid s)) \leq 0$ since $\pi(a \mid s) \leq 1$, we get that

$$\sum_{t=1}^T \mathbb{E}_{a \sim \pi^e(s)}\left[\bar{f}^t(s,a)\right] \leq \frac{1}{\eta}\mathbb{E}_{a \sim \pi^e(s)}\left[\log \pi^{T+1}(a \mid s) - \log \pi^1(a \mid s)\right] + \frac{\eta T}{(1-\gamma)^2}$$

$$\leq -\frac{1}{\eta}\mathbb{E}_{a \sim \pi^e(s)}\left[\log \pi^1(a \mid s)\right] + \frac{\eta T}{(1-\gamma)^2}.$$

Using the fact that $\pi^1(s) = \text{Uniform}(\mathcal{A})$, we get that

$$\sum_{t=1}^T \mathbb{E}_{a \sim \pi^e(s)}\left[\bar{f}^t(s,a)\right] \leq \frac{\log(A)}{\eta} + \frac{\eta T}{(1-\gamma)^2}.$$

Setting $\eta = (1-\gamma)\sqrt{\log(A)/T}$,

$$\sum_{t=1}^T \mathbb{E}_{a \sim \pi^e(s)}\left[\bar{f}^t(s,a)\right] \leq \frac{2}{1-\gamma}\sqrt{\log(A)T}.$$

The final bound follows by taking expectation on both the sides w.r.t. $s \sim d^{\pi^e}$.   $\square$

### C.1.2 Hybrid Analysis Under Approximate Bellman Completeness

Let $f^t$ be the output of Algorithm 2, on policy $\pi^t$ at round $t$ of Algorithm 1. An application of Lemma 3 for each $(\pi_t, f_t)$ implies that

$$\sum_{t=1}^{T} V^{\pi^e} - V^{\pi^t} \le \sum_{t=1}^{T} \mathbb{E}_{s_0, a_0 \sim \mu_0} \Big[ |f^t(s_0, a_0)) - Q^{\pi^t}(s_0, a_0)| \Big]$$

$$+ \frac{1}{1-\gamma} \sum_{t=1}^{T} \mathbb{E}_{s, a \sim d^{\pi^e}} \Big[ \Big( \mathcal{T}^{\pi^t} f^t(s, a) - f^t(s, a) \Big) \Big]$$

$$+ \frac{1}{1-\gamma} \sum_{t=1}^{T} \mathbb{E}_{s, a \sim d^{\pi^e}} \big[ f^t(s, a) - f^t(s, \pi^t(s)) \big],$$

We bound each of the terms on the RHS above separately below:

- *Term 1:* We start by noting that

$$\sum_{t=1}^{T} \mathbb{E}_{s, a \sim \mu_0} \Big[ |f^t(s, \pi^t(s)) - Q^{\pi^t}(s, \pi^t(s))| \Big] = \sum_{t=1}^{T} \|f^t - Q^{\pi^t}\|_{1, \mu_0}$$

$$\le \sum_{t=1}^{T} \|f^t - Q^{\pi^t}\|_{2, \mu_0}$$

$$\le \sum_{t=1}^{T} \sqrt{\frac{1}{1-\gamma}} \Big\| f^t - Q^{\pi^t} \Big\|_{2, d^{\pi^t}},$$

  where the first inequality is due to Jensen's inequality, and the second inequality is from Lemma 2. Using Lemma 5, we get

$$\sum_{t=1}^{T} \mathbb{E}_{s \sim \mu_0} \Big[ |f^t(s, \pi^t(s)) - V^{\pi^t}(s)| \Big] \le T \sqrt{\frac{\Delta_{\text{on}}}{1-\gamma}}.$$

- *Term 2:* Using offline coverage in Definition 3, we get that

$$\sum_{t=1}^{T} \mathbb{E}_{s, a \sim d^{\pi^e}} \Big[ \Big( \mathcal{T}^{\pi^t} f^t(s, a) - f^t(s, a) \Big) \Big] \le C_{\text{off}, \pi^e} \sum_{t=1}^{T} \Big\| f^t - \mathcal{T}^{\pi^t} f^t \Big\|_{2, \nu}$$

$$\le C_{\text{off}, \pi^e} T \sqrt{\Delta_{\text{off}}},$$

  where the last line follows from the bound in Lemma 5.

- *Term 3:* Lemma 6 implies that

$$\sum_{t=1}^{T} \mathbb{E}_{s \sim d_h^{\pi^e}} \big[ f^t(s, \pi^e(s)) - f^t(s, \pi^t(s)) \big] \le \frac{2}{1-\gamma} \sqrt{\log(A) T}.$$

Combining the above bound, we get that

$$\sum_{t=1}^{T} \mathbb{E}_{s \sim \mu_0} \Big[ V^{\pi^e}(s) - V^{\pi^t}(s) \Big] \le T \sqrt{\frac{\Delta_{\text{on}}}{1-\gamma}} + \frac{C_{\text{off}, \pi^e} T}{1-\gamma} \sqrt{\Delta_{\text{off}}} + \frac{2}{(1-\gamma)^2} \sqrt{\log(A) T}. \quad (22)$$

### C.1.3 Natural Policy Gradient Analysis (Without Bellman Completeness)

Let $\pi^t$ and $f^t$ be the corresponding policies and value functions at round $t$, and recall the definition $\bar{f}^t(s, a) = f^t(s, a) - f^t(s, \pi^t(s))$. Using Lemma 1, we get that

$$\mathbb{E}_{s \sim \mu_0} [V^{\pi^e}(s) - V^{\pi^t}(s)]$$

$$= \frac{1}{1-\gamma} \mathbb{E}_{s, a \sim d^{\pi^e}} [A^{\pi^t}(s, a)]$$

$$\leq \frac{1}{1-\gamma} \mathbb{E}_{s,a\sim d^{\pi^e}}[\bar{f}^t(s,a)] + \frac{1}{1-\gamma}\sqrt{\mathbb{E}_{s,a\sim d^{\pi^e}}[(\bar{f}^t(s,a) - A^{\pi^t}(s,a))^2]}$$

$$\leq \frac{1}{1-\gamma} \mathbb{E}_{s,a\sim d^{\pi^e}}[\bar{f}^t(s,a)] + \frac{1}{1-\gamma}\sqrt{C_{\mathrm{npg},\pi^e}\mathbb{E}_{s,a\sim d^{\pi^t}}[(\bar{f}^t(s,a) - A^{\pi^t}(s,a))^2]},$$
(23)

where the second-last line above follows from Jensen's inequality, and the last line is by invoking Definition 2. We next bound the second term in the right hand side. Using Lemma 2-(a), we get that

$$\mathbb{E}_{s,a\sim d^{\pi^t}}[(\bar{f}^t(s,a) - A^{\pi^t}(s,a))^2] = \mathbb{E}_{s,a\sim d^{\pi^t}}[(f^t(s,a) - f^t(s,\pi^t(s)) - Q^{\pi^t}(s,a) + Q^{\pi^t}(s,\pi^t(s)))^2]$$

$$\leq \mathbb{E}_{s,a\sim d^{\pi^t}}[2(f^t(s,a) - Q^{\pi^t}(s,a))^2 + 2(Q^{\pi^t}(s,\pi^t(s)) - f^t(s,\pi^t(s)))^2]$$

$$\leq \mathbb{E}_{s,a\sim d^{\pi^t}}[2(f^t(s,a) - Q^{\pi^t}(s,a))^2 + 2(\mathbb{E}_{a'\sim\pi^t(s)}f^t(s,a') - Q^{\pi^t}(s,a'))^2]$$

$$\leq 4\mathbb{E}_{s,a\sim d^{\pi^t}}[(f^t(s,a) - Q^{\pi^t}(s,a))^2]$$

$$\leq 4\Delta_{\mathrm{on}},$$

where the second last line follows from Jensen's inequality, and the last line uses the bound from Lemma 5. Plugging the above in (23), we get that

$$\mathbb{E}_{s\sim\mu_0}[V^{\pi^e}(s) - V^{\pi^t}(s)] \leq \frac{1}{1-\gamma}\mathbb{E}_{s,a\sim d^{\pi^e}}[\bar{f}^t(s,a)] + \frac{2}{(1-\gamma)}\sqrt{C_{\mathrm{npg},\pi^e}\Delta_{\mathrm{on}}}.$$

Summing the above expression for all $t \in [T]$, we get

$$\sum_{t=1}^{T} V^{\pi^e} - V^{\pi^t} \leq \frac{1}{1-\gamma}\sum_{t=1}^{T}\mathbb{E}_{s,a\sim d^{\pi^e}}[\bar{f}^t(s,a)] + \frac{2T}{(1-\gamma)}\sqrt{C_{\mathrm{npg},\pi^e}\Delta_{\mathrm{on}}}.$$

Plugging the bound from Lemma 6 in the above, we get that

$$\sum_{t=1}^{T} V^{\pi^e} - V^{\pi^t} \leq \frac{2}{(1-\gamma)^2}\sqrt{\log(A)T} + \frac{2T}{(1-\gamma)}\sqrt{C_{\mathrm{npg},\pi^e}\Delta_{\mathrm{on}}}.$$
(24)

### C.1.4 FINAL BOUND ON CUMULATIVE SUBOPTIMALITY

Combining the bounds from (22) and (24), we get that

$$\sum_{t=1}^{T} V^{\pi^e} - V^{\pi^t} \leq \min\left\{ \underbrace{\frac{2}{(1-\gamma)^2}\sqrt{\log(A)T} + \frac{2T}{(1-\gamma)}\sqrt{C_{\mathrm{npg},\pi^e}\Delta_{\mathrm{on}}},}_{(a)} \right.$$

$$\left. \underbrace{T\sqrt{\frac{\Delta_{\mathrm{on}}}{1-\gamma}} + \frac{2}{(1-\gamma)^2}\sqrt{\log(A)T} + \frac{C_{\mathrm{off},\pi^e}T}{(1-\gamma)}\sqrt{\Delta_{\mathrm{off}}}}_{(b)} \right\},$$
(25)

where, from Lemma 5, recall that

$$\Delta_{\mathrm{on}} = \frac{12}{(1-\gamma)^2}\min\left\{\frac{1}{\lambda},\varepsilon_{\mathrm{be}}\right\} + \frac{12\varepsilon_{\mathrm{stat}}}{\lambda(1-\gamma)},$$

$$\Delta_{\mathrm{off}} = \frac{40}{(1-\gamma)}\left(\frac{\lambda\varepsilon_{\mathrm{be}}}{1-\gamma} + \varepsilon_{\mathrm{stat}}\right).$$
(26)

Plugging in the bounds on $\Delta_{\mathrm{on}}$ and $\Delta_{\mathrm{off}}$ in (25), we get that

$$(a) \leq \frac{2}{(1-\gamma)^2}\sqrt{\log(A)T} + \frac{8T}{(1-\gamma)^2}\sqrt{C_{\mathrm{npg},\pi^e}\min\left\{\frac{1}{\lambda},\varepsilon_{\mathrm{be}}\right\}} + \frac{8T}{(1-\gamma)^{3/2}}\sqrt{\frac{C_{\mathrm{npg},\pi^e}\varepsilon_{\mathrm{stat}}}{\lambda}},$$

and,

$$(b) \leq \frac{4T}{(1-\gamma)^{3/2}}\sqrt{\min\left\{\frac{1}{\lambda},\varepsilon_{\mathrm{be}}\right\}} + \frac{4T\sqrt{\varepsilon_{\mathrm{stat}}}}{\lambda(1-\gamma)} + \frac{2}{(1-\gamma)^2}\sqrt{\log(A)T} + \frac{7TC_{\mathrm{off},\pi^e}}{(1-\gamma)^2}\sqrt{\lambda\varepsilon_{\mathrm{be}}} + \frac{7TC_{\mathrm{off},\pi^e}}{(1-\gamma)^{3/2}}\sqrt{\varepsilon_{\mathrm{stat}}}.$$

Note that $\lambda$ is a free parameter in the above, which is chosen by the algorithm. We provide an upper bound on the cumulative suboptimality under two separate cases (we set a different value of $\lambda$, and get a different bound on $\varepsilon_{\text{stat}}$ in the two cases):

- *Case 1:* $\varepsilon_{\text{be}} \leq 1/T$: In this case, we set $\lambda = 1$. Thus, from the bound in (7), we get that $\varepsilon_{\text{stat}} \leq 128/T$, which implies that

$$(a) \leq \frac{2}{(1-\gamma)^2}\sqrt{\log(A)T} + \frac{8T}{(1-\gamma)^2}\sqrt{C_{\text{npg},\pi^e}\varepsilon_{\text{be}}} + \frac{2T}{(1-\gamma)^{3/2}}\sqrt{C_{\text{npg},\pi^e}\varepsilon_{\text{stat}}}$$

$$\leq \frac{2}{(1-\gamma)^2}\sqrt{\log(A)T} + \frac{30}{(1-\gamma)^2}\sqrt{C_{\text{npg},\pi^e}T},$$

where the last line follows from the fact that $\varepsilon_{\text{be}} \leq 1/T$ and $\varepsilon_{\text{stat}} \leq 128/T$. Additionally, we also have that

$$(b) \leq \frac{4T}{(1-\gamma)^{3/2}}\sqrt{\varepsilon_{\text{be}}} + \frac{4T\sqrt{\varepsilon_{\text{stat}}}}{(1-\gamma)} + \frac{2}{(1-\gamma)^2}\sqrt{\log(A)T} + \frac{7TC_{\text{off},\pi^e}}{(1-\gamma)^2}\sqrt{\varepsilon_{\text{be}}} + \frac{7TC_{\text{off},\pi^e}}{(1-\gamma)^{3/2}}\sqrt{\varepsilon_{\text{stat}}}$$

$$\leq \frac{52\sqrt{T}}{(1-\gamma)^{3/2}} + \frac{2}{(1-\gamma)^2}\sqrt{\log(A)T} + \frac{91}{(1-\gamma)^2}\sqrt{C_{\text{off},\pi^e}^2 T}$$

$$\leq \frac{60}{(1-\gamma)^2}\sqrt{\log(A)T} + \frac{100}{(1-\gamma)^2}\sqrt{C_{\text{off},\pi^e}^2 T},$$

where the second-last line uses the fact that $\varepsilon_{\text{be}} \leq 1/T$, and the last line holds since $1/(1-\gamma) \geq 1$ and $\log(A) \geq 1$.

Plugging the above bounds in (25), we get

$$\sum_{t=1}^{T} V^{\pi^e} - V^{\pi^t} \leq \frac{60}{(1-\gamma)^2}\sqrt{\log(A)T} + \frac{100}{(1-\gamma)^2}\sqrt{\min\left\{C_{\text{npg},\pi^e}, C_{\text{off},\pi^e}^2\right\}\cdot T}.$$

- *Case 2:* $\varepsilon_{\text{be}} > 1/T$: In this case, we set $\lambda = T/2 - 1$. Thus, from the bound in (7), we get that $\varepsilon_{\text{stat}} = 64$, which implies that

$$(a) \leq \frac{2}{(1-\gamma)^2}\sqrt{\log(A)T} + \frac{72}{(1-\gamma)^2}\sqrt{C_{\text{npg},\pi^e}T}.$$

Plugging the above bounds in (25), we get

$$\sum_{t=1}^{T} V^{\pi^e} - V^{\pi^t} \leq \frac{2}{(1-\gamma)^2}\sqrt{\log(A)T} + \frac{72}{(1-\gamma)^2}\sqrt{C_{\text{npg},\pi^e}T}.$$

### C.1.5 SAMPLE COMPLEXITY BOUND

Let $n_{\text{on}}$ and $n_{\text{off}}$ denote the total number of on-policy online sample, and offline samples, collected by Algorithm 2. In the following, we give a bound on $n_{\text{on}}$ and $n_{\text{off}}$ for finding a $\varepsilon$-suboptimal policy.

**Corollary 1.** *Consider the setting of Theorem 1. Then, in order to guarantee that the returned policy $\widehat{\pi}$ is $\varepsilon$-suboptimal w.r.t. to the optimal policy $\pi^\star$ (for the underlying MDP), the number of sampled offline and online samples required by HAC in Algorithm 1 is given by:*

- *Under approximate Bellman Complete (when $\varepsilon_{\text{be}} \leq 1/T$):*

$$n_{\text{on}} = n_{\text{off}} = O\left(\frac{(\log(A) + \min\{C_{\text{npg},\pi^\star}, C_{\text{off},\pi^\star}^2\})^3}{\varepsilon^6(1-\gamma)^{14}} \cdot \log(2|\mathcal{F}|/\delta)\right).$$

- *Without Bellman Completeness (when $\varepsilon_{\text{be}} > 1/T$):*

$$n_{\text{off}} = n_{\text{on}} = O\left(\frac{(\log(A) + C_{\text{npg},\pi^\star})^3}{\varepsilon^6(1-\gamma)^{14}} \cdot \log(2|\mathcal{F}|/\delta)\right).$$

We next provide a sample complexity bound. Let $T \geq 4\log(1/\gamma)$ Then, the total number of online samples collected in $T$ rounds of interaction is given by

$$T \cdot K_2 \cdot m_{\text{on}} \lesssim \frac{T^3 \log(2|\mathcal{F}|/\delta)}{(1-\gamma)^2}. \tag{27}$$

Similarly, the total number of offline samples from $\nu$ is given by

$$T \cdot K_2 \cdot m_{\text{off}} \lesssim \frac{T^3 \log(2|\mathcal{F}|/\delta)}{(1-\gamma)^2}. \tag{28}$$

Let $\widehat{\pi} = \text{Uniform}\{(\pi^t)_{t=1}^T\}$, and suppose $\pi^\star$ denote the optimal policy for the underlying MDP. In the following, we provide a bound on total number of samples queried to ensure that $\widehat{\pi}$ is $\varepsilon$-suboptimal. We consider the two cases:

- Under approximate Bellman Complete (when $\varepsilon_{\text{be}} \leq 1/T$):

$$\mathbb{E}\left[V^{\pi^\star} - V^{\widehat{\pi}}\right] \leq \frac{1}{T} \sum_{t=1}^T \mathbb{E}\left[V^{\pi^\star} - V^{\pi^t}\right]$$

$$\leq \frac{60}{(1-\gamma)^2}\sqrt{\frac{\log(A)}{T}} + \frac{100}{(1-\gamma)^2}\sqrt{\frac{\min\left\{C_{\text{npg},\pi^e}, C_{\text{off},\pi^e}^2\right\}}{T}}.$$

  Thus, to ensure that $\mathbb{E}\left[V^{\pi^\star} - V^{\widehat{\pi}}\right] \leq \varepsilon$, we set

$$T = \frac{200}{\varepsilon^2(1-\gamma)^4}\left(36\log(A) + 100\min\left\{C_{\text{npg},\pi^\star}, C_{\text{off},\pi^\star}^2\right\}\right).$$

  This implies a total number of online samples, as

$$O\left(\frac{(\log(A) + \min\{C_{\text{npg},\pi^\star}, C_{\text{off},\pi^\star}^2\})^3}{\varepsilon^6(1-\gamma)^{14}} \cdot \log(2|\mathcal{F}|/\delta)\right).$$

  Total number of offline samples used is the same as above.

- Without Bellman Completeness (when $\varepsilon_{\text{be}} > 1/T$):

$$\mathbb{E}\left[V^{\pi^\star} - V^{\widehat{\pi}}\right] \leq \frac{1}{T} \sum_{t=1}^T \mathbb{E}\left[V^{\pi^\star} - V^{\pi^t}\right]$$

$$\leq \frac{2}{(1-\gamma)^2}\sqrt{\frac{\log(A)}{T}} + \frac{72}{(1-\gamma)^2}\sqrt{\frac{C_{\text{npg},\pi^\star}}{T}}.$$

  Thus, to ensure that $\mathbb{E}\left[V^{\pi^\star} - V^{\widehat{\pi}}\right] \leq \varepsilon$, we set

$$T = O\left(\frac{1}{\varepsilon^2}\left(\frac{\log(A)}{(1-\gamma)^4} + \frac{C_{\text{npg},\pi^\star}}{(1-\gamma)^4}\right)\right).$$

  This implies a total number of online samples, as

$$O\left(\frac{\log(2|\mathcal{F}|/\delta)}{\varepsilon^6(1-\gamma)^{14}}(\log(A) + C_{\text{npg},\pi^\star})^3\right).$$

  Total number of offline samples used is the same as above.

# D    HYBRID POLICY GRADIENT WITH PARAMETERIZED POLICY CLASSES

The previous section uses the softmax policy updates, which may be difficult to handle in applications where action space is continuous. In this section, we present the analysis for HNPG (Algorithm 3) that can work with any differentiably parameterized policy class, including neural network-based policies. Particularly, we consider a parameterized policy class $\Pi = \{\pi_\theta \mid \theta \in \Theta\}$, where $\theta$ is the parameter, which satisfies the following assumption.

**Assumption 2** (Smoothness). *For any parameter $\theta$, state $s$, and action $a$, the function $\ln \pi_\theta(a|s)$ is $\beta$-smooth with respect to $\theta$, i.e.,*

$$|\nabla \ln \pi_\theta(a|s) - \nabla \ln \pi_{\theta'}(a|s)| \leq \beta \|\theta - \theta'\|_2.$$

This smoothness assumption is commonly used in the analysis of NPG style algorithms (Kakade, 2001; Agarwal et al., 2021). Note that vanilla on-policy NPG can be understood as an actor-critic algorithm with compatible function approximation. More formally, on-policy NPG can be understood as first fitting critic with linear function $w^\top \nabla \ln \pi_\theta(a|s)$, i.e. computing $\hat{w} = \arg\min_w \mathbb{E}_{s,a\sim(\nu+\lambda d^{\pi_\theta})} \left(w^\top \nabla \ln \pi_\theta(a|s) - A^{\pi_\theta}(s,a)\right)^2$, followed up a parameter update $\theta' = \theta + \eta \hat{w}$. Our hybrid approach is inspired by this actor-critic interpretation of NPG. As shown in Algorithm 3, every iteration, given $f^t$ which is learned via HPE to approximate $Q^{\pi_{\theta^t}}$, we fit the linear critic $(w^t)^\top \nabla \ln \pi_{\theta^t}(a|s)$ *under both offline and online data* from $\nu$ and $d^{\pi_{\theta^t}}$ respectively. After computing $w^t$, we simply update $\theta^{t+1} = \theta^t + \eta w^t$.

We now illustrate that a similar best-of-both-worlds type of performance guarantee can also be achieved for learning with parameterized policies. We first introduce an assumption which is basically saying that the linear critic $(w^t)^\top \nabla_\theta \ln \pi_\theta(a|s)$ can approximate $f(s,a) - \mathbb{E}_{a\sim\pi(\cdot|s)}f(s,a)$ which itself is used for approximating the advantage $A^\pi(s,a)$ (recall $f$ is used to approximate $Q^\pi$).

**Assumption 3** (Realizability of w). *We assume realizability of the implicit value function in the set $\mathcal{W} = \{w \in \mathbb{R}^d \mid \|w\| \leq W\}$, i.e. for every $\pi \in \Pi$ and $f \in \mathcal{F}$, there exists a $w \in \mathcal{W}$ such that*

$$\sup_{s,a} \|w^\top \nabla_\theta \ln \pi_\theta(a|s) - f(s,a) - \mathbb{E}_{a'\sim\pi(s)} f(s,a')\| = 0.$$

We can relax the above assumption to only hold approximately, but we skip this extension for the sake of conciseness. The next assumption is on the parameterized policy.

**Assumption 4** (Well-parameterized policy class). *For any $\pi_\theta \in \Pi$, we have $\mathbb{E}_{a\sim\pi_\theta}[\nabla_\theta \ln \pi(a|s)] = 0$ for any $s \in \mathcal{S}$.*

This assumption is quite standard and holds for most of the parameterization. For instance, as long as $\pi_\theta(a|s) \propto f_\theta(s,a)$ for some parameterized function $f_\theta$, this condition will hold. Special cases include Gaussian policy $\pi_\theta(\cdot|s) = \mathcal{N}(\mu_\theta(s), \sigma^2 I)$, and flow-based policy parameterization where $a = f_\theta(s,\varepsilon)$ and $\varepsilon \sim \mathcal{N}(0,I)$. Note that Gaussian policy and flow-based policy are commonly used policy parameterizations in practice (examples include TRPO Schulman et al. (2015), PPO Schulman et al. (2017), SAC Haarnoja et al. (2018), etc.).

Finally, for the sake of simplicity, our bounds in this section depend on the concentrability coefficient, which we define below.

**Definition 4** (Concentrability coefficient). *Given the offline distribution $\nu$, for any policy $\pi^e$, we define the concentrability coefficient as*

$$\bar{C}_{\text{off},\pi^e} := \sup_{s,a} \frac{d^{\pi^e}(s,a)}{\nu(s,a)}.$$

Clearly, bounded concentrability coefficient implies bounded Bellman error transfer coefficient (Definition 3), but the converse does not hold. Our main result in this section is the following bound for HNPG algorithm (given in Algorithm 3) that holds for parameterized policy classes.

**Theorem 2** (Cumulative suboptimality). *Fix any $\delta \in (0,1)$, and let $\nu$ be an offline data distribution. Suppose Assumption 1, 2, 3 and 4 hold for the function class $\mathcal{F}$, policy class $\Pi$ and the critic class $\mathcal{W}$. Additionally, suppose that the subroutine HPE is run with parameters $K_1 = 4\lceil\log(1/\gamma)\rceil$, $K_2 = K_1 + T$, and $m_{\text{off}} = m_{\text{on}} = \frac{2T \log(2\max\{|\mathcal{F}|,(W/T)^d\}/\delta)}{(1-\gamma)^2}$. Then, with probability at least $1-\delta$, HNPG satisfies the following bounds on cumulative suboptimality w.r.t. any comparator policy $\pi^e$:*

- *Under approximate Bellman Complete (when $\varepsilon_{\mathrm{be}} \leq 1/T$):*

$$\sum_{t=1}^{T} V^{\pi^e} - V^{\pi_{\theta^t}} \leq \mathcal{O}\left(\frac{1}{1-\gamma}\sqrt{\beta W^2 \log(A) T} + \frac{1}{(1-\gamma)^2}\sqrt{\min\{C_{\mathrm{npg},\pi^e}, \bar{C}_{\mathrm{off},\pi^e}\} \cdot T}\right).$$

- *Without Bellman Completeness (when $\varepsilon_{\mathrm{be}} > 1/T$):*

$$\sum_{t=1}^{T} V^{\pi^e} - V^{\pi_{\theta^t}} \leq \mathcal{O}\left(\frac{1}{1-\gamma}\sqrt{\beta W^2 \log(A) T} + \frac{1}{(1-\gamma)^2}\sqrt{C_{\mathrm{npg},\pi^e} T}\right).$$

*where $\pi_{\theta^t}$ denotes the policy at round $t$.*

Thus, HNPG exhibits a best-of-best-worlds behavior in the sense that it can operate with/without approximate Bellman Completeness, and in both cases enjoys a meaningful cumulative sub-optimality bound.

### D.1 UPDATE RULE

We first recall the update rule. In Algorithm 3, we run Algorithm 2 to get an estimate $f^t$ corresponding to the value function for $\pi^t$. With a fitted $f^t$, we then estimate a linear critic to approximate the advantage on both the offline and online data. In particular, we fit $w^t$ on top of the feature $\phi^t(s,a) := \nabla \log \pi_\theta(a|s)|_{\theta=\theta^t}$ such that

$$w^t = \underset{w}{\mathrm{argmin}} \, \widehat{\mathbb{E}}_{(s,a)\sim\mathcal{D}_{\mathrm{off}}}\left[\left(w^\top \phi^t(s,a) - f^t(s,a) + \mathbb{E}_{a'\sim\pi_{\theta^t}(s)}[f^t(s,a')]\right)^2\right]$$
$$+ \lambda \widehat{\mathbb{E}}_{(s,a)\sim\mathcal{D}_{\mathrm{on}}}\left[\left(w^\top \phi^t(s,a) - f^t(s,a) + \mathbb{E}_{a'\sim\pi_{\theta^t}(s)}[f^t(s,a')]\right)^2\right]. \tag{29}$$

Once we have compute $w^t$, our policy update procedure is defined as follows:
$$\theta^{t+1} := \theta^t + \eta w^t.$$

We next provide a generalization bound for the above. Let

$$\widehat{L}^t(w) = \widehat{\mathbb{E}}_{\mathcal{D}_{\mathrm{off}}}\left[\left(w^\top \phi^t(s,a) - f^t(s,a) + \mathbb{E}_{a'\sim\pi_{\theta^t}(s)}[f^t(s,a')]\right)^2\right]$$
$$+ \lambda \widehat{\mathbb{E}}_{\mathcal{D}_{\mathrm{on}}}\left[\left(w^\top \phi^t(s,a) - f^t(s,a) + \mathbb{E}_{a'\sim\pi_{\theta^t}(s)}[f^t(s,a')]\right)^2\right],$$

and its population counterpart

$$L^t(w) = \mathbb{E}_{(s,a)\sim\nu}\left[\left(w^\top \phi^t(s,a) - f^t(s,a) + \mathbb{E}_{a'\sim\pi_{\theta^t}(s)}[f^t(s,a')]\right)^2\right]$$
$$+ \lambda \mathbb{E}_{(s,a)\sim d^{\pi^t}}\left[\left(w^\top \phi^t(s,a) - f^t(s,a) + \mathbb{E}_{a'\sim\pi_{\theta^t}(s)}[f^t(s,a')]\right)^2\right],$$

where $\phi^t(s,a) := \nabla \log \pi_\theta(a|s)|_{\theta=\theta^t}$. Next, without any loss of generality, assume that $|w^\top \phi^T(s,a)| \leq 1/1-\gamma$ for all $t$ and $s,a$ (This condition can be easily relaxed since $\|w\| \leq W$, and for any $t \geq 1$, Assumption 2 implies that $\|\nabla \log \pi_\theta(a|s)|_{\theta=\theta^t}\| \leq \|\nabla \log \pi_\theta(a|s)|_{\theta=\theta^1}\| + \sum_{s=2}^{T}\|\theta^s - \theta^{s-1}\| \leq \|\nabla \log \pi_\theta(a|s)|_{\theta=\theta^1}\| + \eta t W$). Thus, an application of Lemma 4 implies that the least squares solution $w^t$ satisfies

$$L^t(w^t) \leq \inf_w L^t(w) + \frac{256(1+\lambda)}{(1-\gamma)^2}\frac{\log(2|\mathcal{W}|/\delta)}{\min\{m_{\mathrm{on}}, m_{\mathrm{off}}\}} \leq \inf_w L^t(w) + \underbrace{\frac{256(1+\lambda)}{(1-\gamma)^2}\frac{\log(2(W/T)^d/\delta)}{\min\{m_{\mathrm{on}}, m_{\mathrm{off}}\}}}_{=:\Delta_w}, \tag{30}$$

where the second line follows from a straightforward covering argument of the set $\mathcal{W} = \{w \in \mathbb{R}^d \mid \|w\| \leq W\}$ at scale $1/T$. Using Assumption 3 in the above bound, we get that

$$L^t(w^t) \leq \Delta_w,$$

which implies that

$$\mathbb{E}_{s,a\sim\nu}\left[\left((w^t)^\top \phi^t(s,a) - f^t(s,a) + \mathbb{E}_{a'\sim\pi_{\theta^t}(s)}[f^t(s,a')]\right)^2\right] \leq \Delta_w. \tag{31}$$

and
$$\mathbb{E}_{s,a\sim d^{\pi_{\theta^t}}}\left[\left((w^t)^\top \phi^t(s,a) - f^t(s,a) + \mathbb{E}_{a'\sim\pi_{\theta^t}(s)}[f^t(s,a')]\right)^2\right] \le \frac{\Delta_w}{\lambda}. \tag{32}$$

Next, in order to simply the notation in the following proof, we increase the value of $\varepsilon_{\text{stat}}$ to
$$\varepsilon_{\text{stat}} = 256 \frac{1+\lambda}{(1-\gamma)^2} \cdot \frac{\ln(2\max\{|\mathcal{F}|, (W/T)^d\}/\delta)}{\min\{m_{\text{on}}, m_{\text{off}}\}} \le 256 \frac{1+\lambda}{2T}, \tag{33}$$
which ensures that $\Delta_w \le \varepsilon_{\text{stat}}$.

### D.2 PROOF OF THEOREM 2

#### D.2.1 SUPPORTING TECHNICAL RESULTS

Fix any $t \le T$, and let $\pi_{\theta^t}$ be the policy at round $t$, and $f^t$ be the corresponding value function that is computed using Algorithm 2 at round $t$. We note that an application of Lemma 5 implies that
$$\mathbb{E}_{s,a\sim d^{\pi_{\theta^t}}}[(f^t(s,a) - Q^{\pi_{\theta^t}}(s,a))^2] \le \frac{12}{(1-\gamma)^2}\min\left\{\frac{1}{\lambda}, \varepsilon_{\text{be}}\right\} + \frac{12\varepsilon_{\text{stat}}}{\lambda(1-\gamma)} =: \Delta_{\text{on}} \tag{34}$$
and that,
$$\mathbb{E}_{s,a\sim\nu}[(f^t(s,a) - \mathcal{T}^{\pi_{\theta^t}}f^t(s,a))^2] \le \frac{40}{(1-\gamma)}\left(\frac{\lambda\varepsilon_{\text{be}}}{1-\gamma} + \varepsilon_{\text{stat}}\right) =: \Delta_{\text{off}}. \tag{35}$$

Furthermore, from (31) and (32) recall that
$$\mathbb{E}_{s,a\sim\nu}\left[\left((w^t)^\top\phi^t(s,a) - f^t(s,a) + \mathbb{E}_{a'\sim\pi_{\theta^t}(s)}[f^t(s,a')]\right)^2\right] \le \Delta_w. \tag{36}$$
and that
$$\mathbb{E}_{s,a\sim d^{\pi_{\theta^t}}}\left[\left((w^t)^\top\phi^t(s,a) - f^t(s,a) + \mathbb{E}_{a'\sim\pi_{\theta^t}(s)}[f^t(s,a')]\right)^2\right] \le \frac{\Delta_w}{\lambda}, \tag{37}$$
and that $\Delta_w \le \varepsilon_{\text{stat}}$. Additionally, define the function $g^t$ such that for all $s,a$:
$$g^t(s,a) = \mathbb{E}_{a'\sim\pi_{\theta^t}(s)}[f^t(s,a')] + (w^t)^\top\phi^t(s,a). \tag{38}$$
Before we move to the bound on total suboptimality, we first prove two technical results for $g^t$ that will be useful for the rest of the analysis.

- First, note that
$\|g^t - \mathcal{T}^{\pi_{\theta^t}}g^t\|_{2,\nu}^2$
$$= \mathbb{E}_{s,a\sim\nu}\left[\left(g^t(s,a) - r(s,a) - \mathbb{E}_{s'\sim P(\cdot|s,a),a'\sim\pi_{\theta^t}(s')}[g^t(s',a')]\right)^2\right]$$
$$\overset{(i)}{=} \mathbb{E}_{s,a\sim\nu}\left[\left(\mathbb{E}_{a'\sim\pi_{\theta^t}(s)}f^t(s,a') + (w^t)^\top\phi^t(s,a) - r(s,a) - \mathbb{E}_{s'\sim P(\cdot|s,a),a'\sim\pi_{\theta^t}(s')}[f^t(s,a)]\right)^2\right]$$
$$\overset{(ii)}{\le} 2\mathbb{E}_{s,a\sim\nu}\left[\left((w^t)^\top\phi^t(s,a) - f^t(s,a) + \mathbb{E}_{a'\sim\pi_{\theta^t}(s)}[f^t(s,a')]\right)^2\right]$$
$$\qquad + 2\mathbb{E}_{s,a\sim\nu}\left[\left(f^t(s,a) - r(s,a) - \mathbb{E}_{s'\sim P(\cdot|s,a),a'\sim\pi_{\theta^t}(s')}[f^t(s,a)]\right)^2\right]$$
$$\le 2\Delta_w + 2\|f^t - \mathcal{T}^{\pi_{\theta^t}}f^t\|_{2,\nu}^2$$
$$\overset{(iii)}{\le} 2\Delta_w + 2\Delta_{\text{off}},$$
  where $(i)$ uses the fact that $\mathbb{E}_{a'\sim\pi_{\theta^t}(s')}[\phi^t(s',a')] = 0$ for all $s' \in \mathcal{S}$ in the second term. The inequality $(ii)$ holds from splitting the squares. Finally, $(iii)$ follows from (35). Thus,
$$\|g^t - \mathcal{T}^{\pi_{\theta^t}}g^t\|_{2,\nu} \le \sqrt{2\Delta_{\text{off}} + 2\Delta_w}. \tag{39}$$
- Next, note that since $\mathbb{E}_{a\sim\pi(s)}[\phi^t(s,a)] = 0$ for any $s$, we have $\mathbb{E}_{a\sim\pi_{\theta^t}(\cdot|s)}[g^t(s,a)] = \mathbb{E}_{a\sim\pi_{\theta^t}(\cdot|s)}[f^t(s,a)]$. Thus,
$$\mathbb{E}_{s_0,a_0\sim\mu_0}\left[g^t(s_0,a_0) - Q^{\pi_{\theta^t}}(s_0,a_0)\right] = \mathbb{E}_{s_0,a_0\sim\mu_0}\left[f^t(s_0,a_0) - Q^{\pi_{\theta^t}}(s_0,a_0)\right]$$
$$\le \|f^t - Q^{\pi_{\theta^t}}\|_{1,\mu_0}$$

$$\leq \|f^t - Q^{\pi_{\theta^t}}\|_{2,\mu_0}$$

$$\leq \sqrt{\frac{1}{1-\gamma}} \|f^t - Q^{\pi_{\theta^t}}\|_{2,\pi_{\theta^t}}$$

$$\leq \sqrt{\frac{\Delta_{\text{on}}}{1-\gamma}}, \tag{40}$$

where the third last line is from Jensen's inequality, the second last line is from Lemma 2 and the last line is due to (34).

**Lemma 7.** *Consider the update rule in Algorithm 3, and let the function $g^t$ be defined such that $g^t(s,a) = \mathbb{E}_{a' \sim \pi_{\theta^t}(s)}[f^t(s,a')] + (w^t)^\top \phi^t(s,a)$. Then, setting $\eta = \sqrt{\frac{2\log(A)}{\beta W^2 T}}$, we get that*

$$\sum_{t=1}^{T} \mathbb{E}_{s,a \sim d^{\pi^e}} \left[ [g^t(s,a)] - \mathbb{E}_{a \sim \pi_{\theta^t}(s)}[g^t(s,a)] \right] \leq \sqrt{2\beta W^2 \log(A) T}.$$

*Proof of Lemma 7.* For policy optimization part, we will start by leveraging the smoothness of the log of the policy. For any $s, a$, $\beta$-smoothness implies that

$$\log \frac{\pi_{\theta^{t+1}}(a|s)}{\pi_{\theta^t}(a|s)} \geq \nabla_\theta \log \pi_{\theta^t}(a|s) \cdot \left(\theta^{t+1} - \theta^t\right) - \frac{\beta}{2} \left\|\theta^{t+1} - \theta^t\right\|_2^2$$

$$= \eta \nabla_\theta \log \pi_{\theta^t}(a|s) \cdot w^t - \frac{\eta^2 \beta}{2} \left\|w^t\right\|_2^2. \tag{41}$$

Taking expectation on both the sides w.r.t. $a \sim \pi^e(s)$, we have that:

$$\text{KL}(\pi^e(s)\|\pi_{\theta^t}(s)) - \text{KL}(\pi^e(s)\|\pi_{\theta^{t+1}}(s))$$

$$= \mathbb{E}_{a \sim \pi^e(s)}[\log(\pi_{\theta^{t+1}}(a|s)) - \log(\pi_{\theta^t}(a|s))]$$

$$\geq \eta \mathbb{E}_{a \sim \pi^e(s)}[\nabla_\theta \log \pi_{\theta^t}(a|s) \cdot w^t] - \frac{\eta^2 \beta}{2} \left\|w^t\right\|_2^2$$

$$\geq \eta \mathbb{E}_{a \sim \pi^e(s)}[\nabla_\theta \log \pi_{\theta^t}(a|s) \cdot w^t] - \frac{\eta^2 \beta}{2} W^2$$

$$= \eta(\mathbb{E}_{a \sim \pi^e(s)}[\nabla_\theta \log \pi_{\theta^t}(a|s) \cdot w^t] - \mathbb{E}_{a \sim \pi_{\theta^t}(s)}[\nabla_\theta \log \pi_{\theta^t}(a|s) \cdot w^t]) - \frac{\eta^2 \beta}{2} W^2,$$

where the second line above follows from (41), and the last line follows from the fact that $\mathbb{E}_{a \sim \pi_{\theta^t}(s)}[\nabla_\theta \log \pi_{\theta^t}(a|s)] = 0$ for any $s$. Rearranging the terms, and taking expectation w.r.t. $s \sim d^{\pi^e}$, we get that:

$$\mathbb{E}_{s \sim d^{\pi^e}}[\mathbb{E}_{a \sim \pi^e(s)}[\nabla_\theta \log \pi_{\theta^t}(a|s) \cdot w^t] - [\mathbb{E}_{a \sim \pi_{\theta^t}(s)}\nabla_\theta \log \pi_{\theta^t}(a|s) \cdot w^t]]$$

$$\leq \frac{1}{\eta} \mathbb{E}_{s \sim d^{\pi^e}}[(\text{KL}(\pi^e(s)\|\pi_{\theta^t}(s)) - \text{KL}(\pi^e(s)\|\pi_{\theta^{t+1}}(s))] + \frac{\eta\beta}{2} W^2.$$

Next, recall that definition $g^t(s,a) = \mathbb{E}_{a' \sim \pi_{\theta^t}(s)}[f^t(s,a')] + (w^t)^\top \phi^t(s,a)$ where $\phi^t(s,a) = \nabla_\theta \log \pi_{\theta^t}(a|s)$. Using this in the above, we get that

$$\mathbb{E}_{s \sim d^{\pi^e}}\left[\mathbb{E}_{a \sim \pi^e(s)}[g^t(s,a)] - \mathbb{E}_{a \sim \pi_{\theta^t}(s)}[g^t(s,a)]\right]$$

$$\leq \frac{1}{\eta} \mathbb{E}_{s \sim d^{\pi^e}}[(\text{KL}(\pi^e(s)\|\pi_{\theta^t}(s)) - \text{KL}(\pi^e(s)\|\pi_{\theta^{t+1}}(s))] + \frac{\eta\beta}{2} W^2.$$

Summing the above for $t$ from 1 to $T$, we get that:

$$\sum_{t=1}^{T} \mathbb{E}_{s \sim d^{\pi^e}}\left[\mathbb{E}_{a \sim \pi^e(s)}[g^t(s,a)] - \mathbb{E}_{a \sim \pi_{\theta^t}(s)}[g^t(s,a)]\right]$$

$$\leq \frac{1}{\eta} \mathbb{E}_{s \sim d^{\pi^e}}[(\text{KL}(\pi^e(s)\|\pi_{\theta^1}(s)) - \text{KL}(\pi^e(s)\|\pi_{\theta^{T+1}}(s))] + \frac{\eta\beta}{2} W^2 T$$

$$\leq \frac{1}{\eta} \mathbb{E}_{s \sim d^{\pi^e}}[(\text{KL}(\pi^e(s)\|\pi_{\theta^1}(s))] + \frac{\eta\beta}{2} W^2 T.$$

Using the fact that $\pi_{\theta^1}(s) = \text{Uniform}(A)$, we get that

$$\text{KL}(\pi^e(s)\|\pi_{\theta^1}(s)) \le \mathbb{E}_{a\sim\pi^e(s)}[-\log(\pi_{\theta^1}(s))] = \log(A),$$

which implies that

$$\sum_{t=1}^{T}\mathbb{E}_{s\sim d^{\pi^e}}\left[g^t(s,\pi^e(s)) - g^t(s,\pi_{\theta^t}(s))\right] \le \frac{\log(A)}{\eta} + \frac{\eta\beta}{2}W^2 T.$$

Setting $\eta = \sqrt{\frac{2\log(A)}{\beta W^2 T}}$ concludes the proof.

$\square$

### D.2.2 HYBRID ANALYSIS UNDER APPROXIMATE BELLMAN COMPLETENESS

For any $t \le [T]$, invoking Lemma 3 with $\pi = \pi_{\theta^t}$ and $f = g^t$, we get that

$$V^{\pi^e} - V^{\pi_{\theta^t}} \le \mathbb{E}_{s_0,a_0\sim\mu_0}\left[g^t(s_0,a_0) - Q^{\pi_{\theta^t}}(s_0,a_0)\right]$$
$$+ \frac{1}{1-\gamma}\mathbb{E}_{s,a\sim d^{\pi^e}}\left[\mathcal{T}^{\pi_{\theta^t}}g^t(s,a) - g^t(s,a)\right] + \frac{1}{1-\gamma}\mathbb{E}_{s,a\sim d^{\pi^e}}\left[g^t(s,a) - g^t(s,\pi_{\theta^t}(s))\right].$$

Using Jensen's inequality and Definition 4, we get that

$$V^{\pi^e} - V^{\pi_{\theta^t}} \le \mathbb{E}_{s_0,a_0\sim\mu_0}\left[g^t(s_0,a_0) - Q^{\pi_{\theta^t}}(s_0,a_0)\right]$$
$$+ \frac{\bar{C}_{\text{off},\pi^e}}{1-\gamma}\|\mathcal{T}^{\pi_{\theta^t}}g^t(s,a) - g^t(s,a)\|_{2,\nu} + \frac{1}{1-\gamma}\mathbb{E}_{s,a\sim d^{\pi^e}}\left[g^t(s,a) - g^t(s,\pi_{\theta^t}(s))\right].$$

Adding the above bounds for $t$ from $1$ to $T$, we get

$$\sum_{t=1}^{T}V^{\pi^e} - V^{\pi_{\theta^t}} \le \sum_{t=1}^{T}\mathbb{E}_{s_0,a_0\sim\mu_0}\left[g^t(s_0,a_0) - Q^{\pi_{\theta^t}}(s_0,a_0)\right]$$
$$+ \sum_{t=1}^{T}\frac{\bar{C}_{\text{off},\pi^e}}{1-\gamma}\|\mathcal{T}^{\pi_{\theta^t}}g^t(s,a) - g^t(s,a)\|_{2,\nu} + \sum_{t=1}^{T}\frac{1}{1-\gamma}\mathbb{E}_{s,a\sim d^{\pi^e}}\left[g^t(s,a) - g^t(s,\pi_{\theta^t}(s))\right].$$

We bound each of the terms on the RHS above separately below:

- *Term 1:* Using (40), we get that

$$\sum_{t=1}^{T}\mathbb{E}_{s_0,a_0\sim\mu_0}\left[g^t(s_0,a_0) - Q^{\pi_{\theta^t}}(s_0,a_0)\right] \le T\sqrt{\frac{\Delta_{\text{on}}}{(1-\gamma)}}.$$

- *Term 2:* Using (39) , we get that

$$\sum_{t=1}^{T}\bar{C}_{\text{off},\pi^e}\left\|g^t - \mathcal{T}^\pi g^t\right\|_{2,\nu} \le 2\bar{C}_{\text{off},\pi^e}T\sqrt{\Delta_{\text{off}} + \Delta_w}.$$

- *Term 3:* Using Lemma 7, we get that

$$\sum_{t=1}^{T}\mathbb{E}_{s,a\sim d^{\pi^e}}\left[g^t(s,a) - g^t(s,\pi_{\theta^t}(s))\right] \le \sqrt{2\beta W^2 \log(A)T}.$$

Combining the above bounds, we get that

$$\sum_{t=1}^{T}V^{\pi^e} - V^{\pi_{\theta^t}} \le T\sqrt{\frac{\Delta_{\text{on}}}{(1-\gamma)}} + \frac{2\bar{C}_{\text{off},\pi^e}}{1-\gamma}T\sqrt{\Delta_{\text{off}} + \Delta_w} + \frac{1}{1-\gamma}\sqrt{2\beta W^2 \log(A)T}. \quad (42)$$

### D.2.3 NATURAL POLICY GRADIENT ANALYSIS

The following bound follows by repeating a similar analysis as in Appendix C.1.3. Fix any $t \in [T]$, let $\pi_{\theta^t}$ and $f^t$ be the corresponding policies and value functions at round $t$. Further, recall that $g^t(s,a) = \mathbb{E}_{a'\sim\pi_{\theta^t}(s)}[f^t(s,a')] + (w^t)^\top\phi^t(s,a)$ and define $\bar{g}^t(s,a) = g^t(s,a) - g^t(s,\pi_{\theta^t}(s))$.

Using Lemma 1, we get that

$$\mathbb{E}_{s\sim\mu_0}[V^{\pi^e}(s) - V^{\pi_{\theta t}}(s)]$$

$$=\frac{1}{1-\gamma}\mathbb{E}_{s,a\sim d^{\pi^e}}[A^{\pi_{\theta t}}(s,a)]$$

$$\leq\frac{1}{1-\gamma}\mathbb{E}_{s,a\sim d^{\pi^e}}[\bar{g}^t(s,a)] + \frac{1}{1-\gamma}\sqrt{\mathbb{E}_{s,a\sim d^{\pi^e}}[(\bar{g}^t(s,a) - A^{\pi_{\theta t}}(s,a))^2]}$$

$$\leq\frac{1}{1-\gamma}\mathbb{E}_{s,a\sim d^{\pi^e}}[\bar{g}^t(s,a)] + \frac{1}{1-\gamma}\sqrt{C_{\text{npg},\pi^e}\mathbb{E}_{s,a\sim d^{\pi_t}}[(\bar{g}^t(s,a) - A^{\pi_{\theta t}}(s,a))^2]},$$
$$\tag{43}$$

where the second-last line above follows from Jensen's inequality, and the last line is by invoking Definition 2. We next bound the second term in the right hand side. Using the fact that $\pi_t = \pi_{\theta t}$, we get that

$$\mathbb{E}_{s,a\sim d^{\pi_{\theta t}}}[(\bar{g}^t(s,a) - A^{\pi_{\theta t}}(s,a))^2]$$

$$=\mathbb{E}_{s,a\sim d^{\pi_{\theta t}}}[(g^t(s,a) - g^t(s,\pi_{\theta t}(s)) - Q^{\pi_{\theta t}}(s,a) + Q^{\pi_{\theta t}}(s,\pi_{\theta t}(s)))^2]$$

$$\leq\mathbb{E}_{s,a\sim d^{\pi_{\theta t}}}[2(g^t(s,a) - Q^{\pi_{\theta t}}(s,a))^2 + 2(Q^{\pi_{\theta t}}(s,\pi_{\theta t}(s)) - g^t(s,\pi_{\theta t}(s)))^2]$$

$$=\mathbb{E}_{s,a\sim d^{\pi_{\theta t}}}[2(g^t(s,a) - Q^{\pi_{\theta t}}(s,a))^2 + 2(\mathbb{E}_{a'\sim\pi_{\theta t}(s)}[g^t(s,a') - Q^{\pi_{\theta t}}(s,a')])^2]$$

$$\leq 4\mathbb{E}_{s,a\sim d^{\pi_{\theta t}}}[(g^t(s,a) - Q^{\pi_{\theta t}}(s,a))^2],$$

where the first inequality uses $(a+b)^2 \leq 2a^2 + 2b^2$ for any $a, b$, and the second inequality follows from Jensen's inequality. Adding and subtracting $\mathbb{E}_{a'\sim\pi_{\theta t}(s)}[f^t(s,a')])$ inside the expectation, further decomposing the above, and again applying Jensen's inequality, we get that

$$\mathbb{E}_{s,a\sim d^{\pi_{\theta t}}}[(\bar{g}^t(s,a) - A^{\pi_{\theta t}}(s,a))^2] \leq 8\mathbb{E}_{s,a\sim d^{\pi_{\theta t}}}[(g^t(s,a) - \mathbb{E}_{a'\sim\pi_{\theta t}(s)}[f^t(s,a')])^2]$$

$$+ 8\mathbb{E}_{s,a\sim d^{\pi_{\theta t}}}[(f^t(s,a) - Q^{\pi_{\theta t}}(s,a))^2]$$

$$\leq \frac{\Delta_w}{\lambda} + \Delta_{\text{on}},$$

where the last line uses the bound from (37) and Lemma 5. Plugging the above in (43), we get that

$$\mathbb{E}_{s\sim\mu_0}[V^{\pi^e}(s) - V^{\pi_{\theta t}}(s)] \leq \frac{1}{1-\gamma}\mathbb{E}_{s,a\sim d^{\pi^e}}[\bar{g}^t(s,a)] + \frac{4}{(1-\gamma)}\sqrt{\frac{C_{\text{npg},\pi^e}\Delta_w}{\lambda}} + \frac{4}{(1-\gamma)}\sqrt{C_{\text{npg},\pi^e}\Delta_{\text{on}}}.$$

Summing the above expression for all $t \in [T]$ implies that

$$V^{\pi^e} - V^{\pi_{\theta t}} \leq \frac{1}{1-\gamma}\sum_{t=1}^{T}\mathbb{E}_{s,a\sim d^{\pi^e}}[\bar{g}^t(s,a)] + \frac{4T}{(1-\gamma)}\sqrt{\frac{C_{\text{npg},\pi^e}\Delta_w}{\lambda}} + \frac{4T}{(1-\gamma)}\sqrt{C_{\text{npg},\pi^e}\Delta_{\text{on}}}.$$

Using the bound for the first term from Lemma 7 in the above, we get that

$$V^{\pi^e} - V^{\pi_{\theta t}} \leq \frac{1}{1-\gamma}\sqrt{2\beta W^2 \log(A)T} + \frac{4T}{(1-\gamma)}\sqrt{\frac{C_{\text{npg},\pi^e}\Delta_w}{\lambda}} + \frac{4T}{(1-\gamma)}\sqrt{C_{\text{npg},\pi^e}\Delta_{\text{on}}}.$$
$$\tag{44}$$

### D.2.4 FINAL BOUND ON CUMULATIVE SUBOPTIMALITY

Combining the bounds from (42) and (44), we get that

$$\sum_{t=1}^{T}V^{\pi^e} - V^{\pi_{\theta t}} \leq \min\Bigg\{ \underbrace{\frac{1}{1-\gamma}\sqrt{4\beta W^2 \log(A)T} + \frac{2T}{(1-\gamma)}\sqrt{\frac{C_{\text{npg},\pi^e}\Delta_w}{\lambda}} + \frac{4T}{(1-\gamma)}\sqrt{C_{\text{npg},\pi^e}\Delta_{\text{on}}}}_{(a)},$$

$$\underbrace{T\sqrt{\frac{\Delta_{\text{on}}}{(1-\gamma)}} + \frac{2\bar{C}_{\text{off},\pi^e}T}{1-\gamma}\sqrt{\Delta_{\text{off}} + \Delta_w} + \frac{1}{1-\gamma}\sqrt{2\beta W^2 \log(A)T}}_{(b)} \Bigg\},$$
$$\tag{45}$$

where recall that

$$\Delta_{\mathrm{on}} = \frac{12}{(1-\gamma)^2} \min\{\tfrac{1}{\lambda}, \varepsilon_{\mathrm{be}}\} + \frac{12\varepsilon_{\mathrm{stat}}}{\lambda(1-\gamma)}$$

$$\Delta_{\mathrm{off}} = \frac{40}{(1-\gamma)} \left( \frac{\lambda\varepsilon_{\mathrm{be}}}{1-\gamma} + \varepsilon_{\mathrm{stat}} \right)$$

$$\Delta_w = \varepsilon_{\mathrm{stat}},$$

$$m_{\mathrm{off}} = m_{\mathrm{on}} = \frac{2T \log(|\mathcal{F}/\delta|)}{(1-\gamma)}$$

$$\varepsilon_{\mathrm{stat}} \le 256 \frac{1+\lambda}{(1-\gamma)^2} \cdot \frac{\ln(2 \max\{|\mathcal{F}| \wedge |\mathcal{W}|\}/\delta)}{\min\{m_{\mathrm{on}}, m_{\mathrm{off}}\}} \le 256 \frac{1+\lambda}{2T},$$

due to Lemma 5, and (33) and (30). Plugging in the bounds on $\Delta_{\mathrm{on}}$ and $\Delta_{\mathrm{off}}$ in (45), we get that

$$(a) \le \frac{1}{1-\gamma} \sqrt{2\beta W^2 \log(A) T} + \frac{16T}{(1-\gamma)^{3/2}} \sqrt{\frac{C_{\mathrm{npg}, \pi^e} \varepsilon_{\mathrm{stat}}}{\lambda}} + \frac{16T}{(1-\gamma)^2} \sqrt{C_{\mathrm{npg}, \pi^e} \min\left\{ \frac{1}{\lambda}, \varepsilon_{\mathrm{be}} \right\}},$$

and,

$$(b) \le \frac{4T}{(1-\gamma)^{3/2}} \sqrt{\min\left\{ \frac{1}{\lambda}, \varepsilon_{\mathrm{be}} \right\}} + \frac{4T}{(1-\gamma)} \sqrt{\frac{\varepsilon_{\mathrm{stat}}}{\lambda}} + \frac{14T\bar{C}_{\mathrm{off}, \pi^e}}{(1-\gamma)^2} \sqrt{\lambda \varepsilon_{\mathrm{be}}}$$

$$+ \frac{14T\bar{C}_{\mathrm{off}, \pi^e}}{(1-\gamma)^{3/2}} \sqrt{\varepsilon_{\mathrm{stat}}} + \frac{1}{1-\gamma} \sqrt{2\beta W^2 \log(A) T}.$$

Note that $\lambda$ is a free parameter in the above, which is chosen by the algorithm. We provide an upper bound on the cumulative suboptimality under two separate cases (we set a different value of $\lambda$, and get a different bound on $\varepsilon_{\mathrm{stat}}$ in the two cases):

- *Case 1: Under approximate Bellman Complete (when $\varepsilon_{\mathrm{be}} \le 1/T$):* In this case, we set $\lambda = 1$. Thus, from the bound in (7), we get that $\varepsilon_{\mathrm{stat}} \le 256/T$, which implies that

$$(a) \le \frac{1}{1-\gamma} \sqrt{2\beta W^2 \log(A) T} + \frac{16T}{(1-\gamma)^{3/2}} \sqrt{C_{\mathrm{npg}, \pi^e} \varepsilon_{\mathrm{stat}}} + \frac{16T}{(1-\gamma)^2} \sqrt{C_{\mathrm{npg}, \pi^e} \varepsilon_{\mathrm{be}}}$$

$$\le \frac{1}{1-\gamma} \sqrt{2\beta W^2 \log(A) T} + \frac{300}{(1-\gamma)^2} \sqrt{C_{\mathrm{npg}, \pi^e} T},$$

where the last line follows from the fact that $\varepsilon_{\mathrm{be}} \le 1/T$. Additionally, we also have that

$$(b) \le \frac{350}{(1-\gamma)^2} \sqrt{\bar{C}_{\mathrm{off}, \pi^e}^2 T} + \frac{1}{1-\gamma} \sqrt{2\beta W^2 \log(A) T},$$

where the last line uses the fact that $\varepsilon_{\mathrm{be}} \le 1/T$, and that $\bar{C}_{\mathrm{off}, \pi^e} \ge 1$.
Plugging the above bounds in (25), we get

$$\sum_{t=1}^T V^{\pi^e} - V^{\pi_{\theta t}} \le \frac{1}{1-\gamma} \sqrt{2\beta W^2 \log(A) T} + \frac{350}{(1-\gamma)^2} \sqrt{\min\left\{ C_{\mathrm{npg}, \pi^e}, \bar{C}_{\mathrm{off}, \pi^e}^2 \right\} \cdot T}.$$

- *Case 2: Without Bellman Completeness (when $\varepsilon_{\mathrm{be}} > 1/T$)* In this case, we set $\lambda = \frac{T}{2} - 1$. Thus, from the bound in (7), we get that $\varepsilon_{\mathrm{stat}} = 128$, which implies that

$$(a) \le \frac{1}{1-\gamma} \sqrt{\beta W^2 \log(A) T} + \frac{56}{(1-\gamma)^2} \sqrt{C_{\mathrm{npg}, \pi^e} T}.$$

Plugging the above bounds in (25), we get

$$\sum_{t=1}^T V^{\pi^e} - V^{\pi_{\theta t}} \le \frac{1}{1-\gamma} \sqrt{2\beta W^2 \log(A) T} + \frac{56}{(1-\gamma)^2} \sqrt{C_{\mathrm{npg}, \pi^e} T}.$$

### D.2.5 SAMPLE COMPLEXITY BOUND

**Corollary 2** (Sample complexity). *Consider the setting of Theorem 2. Then, in order to guarantee that the returned policy $\widehat{\pi}$ is $\varepsilon$-suboptimal w.r.t. to the optimal policy $\pi^\star$ (for the underlying MDP), the number of sampled offline and online samples required by* HNPG *in Algorithm 3 is given by:*

- *Under approximate Bellman Complete (when $\varepsilon_{\mathrm{be}} \leq 1/T$):*

$$n_{\mathrm{on}} = n_{\mathrm{off}} = O\left( \frac{\left(\beta W^2 \log(A) + \min\{C_{\mathrm{npg},\pi^\star}, \bar{C}^2_{\mathrm{off},\pi^\star}\}/(1-\gamma)^2\right)^3}{\varepsilon^6 (1-\gamma)^8} \cdot \log(2\max\{(W/T)^d, |\mathcal{F}|\}/\delta)\right).$$

- *Without Bellman Completeness (when $\varepsilon_{\mathrm{be}} > 1/T$):*

$$n_{\mathrm{on}} = n_{\mathrm{off}} = O\left( \frac{\left(\beta W^2 \log(A) + C_{\mathrm{npg},\pi^\star}/(1-\gamma)^2\right)^3}{\varepsilon^6 (1-\gamma)^8} \cdot \log(2\max\{(W/T)^d, |\mathcal{F}|\}/\delta)\right).$$

*In particular,* HNPG *draws the same number of offline samples, and on-policy online samples.*

We next provide a sample complexity bound. Let $T \geq 4\log(1/\gamma)$ Then, the total number of online samples collected in $T$ rounds of interaction is given by

$$T \cdot K_2 \cdot m_{\mathrm{on}} \lesssim \frac{T^3 (\log(2\max\{(W/T)^d, |\mathcal{F}|\}/\delta))}{(1-\gamma)^2}. \tag{46}$$

Similarly, the total number of offline samples from $\nu$ is given by

$$T \cdot K_2 \cdot m_{\mathrm{off}} \lesssim \frac{T^3 (\log(2\max\{(W/T)^d, |\mathcal{F}|\}/\delta))}{(1-\gamma)^2}. \tag{47}$$

Let $\widehat{\pi} = \mathrm{Uniform}\{(\pi_{\theta^t})_{t=1}^T\}$, and suppose $\pi^\star$ denote the optimal policy for the underlying MDP. In the following, we provide a bound on total number of samples queried to ensure that $\widehat{\pi}$ is $\varepsilon$-suboptimal. We consider the two cases:

- *Case 1: Under approximate Bellman Complete (when $\varepsilon_{\mathrm{be}} \leq 1/T$):*

$$\mathbb{E}\left[V^{\pi^\star} - V^{\widehat{\pi}}\right] \leq \frac{1}{T}\sum_{t=1}^T \mathbb{E}\left[V^{\pi^\star} - V^{\pi_{\theta^t}}\right]$$

$$\leq \frac{1}{1-\gamma}\sqrt{2\beta W^2 \log(A)T} + \frac{350}{(1-\gamma)^2}\sqrt{\min\left\{C_{\mathrm{npg},\pi^e}, \bar{C}^2_{\mathrm{off},\pi^e}\right\} \cdot T}.$$

Thus, to ensure that $\mathbb{E}\left[V^{\pi^\star} - V^{\widehat{\pi}}\right] \leq \varepsilon$, we set

$$T = O\left(\frac{1}{\varepsilon^2}\left(\frac{\beta W^2 \log(A)}{(1-\gamma)^2} + \frac{1}{(1-\gamma)^4}\min\{C_{\mathrm{npg},\pi^\star}, \bar{C}^2_{\mathrm{off},\pi^\star}\}\right)\right).$$

This implies a total number of online samples, as

$$O\left(\frac{\log(2\max\{(W/T)^d, |\mathcal{F}|\}/\delta)}{\varepsilon^6 (1-\gamma)^8}\left(\beta W^2 \log(A) + \frac{1}{(1-\gamma)^2}\min\{C_{\mathrm{npg},\pi^\star}, \bar{C}^2_{\mathrm{off},\pi^\star}\}\right)^3\right).$$

Total number of offline samples used is the same as above.

- *Case 2: Without Bellman Completeness (when $\varepsilon_{\mathrm{be}} > 1/T$):*

$$\mathbb{E}\left[V^{\pi^\star} - V^{\widehat{\pi}}\right] \leq \frac{1}{T}\sum_{t=1}^T \mathbb{E}\left[V^{\pi^\star} - V^{\pi_{\theta^t}}\right]$$

$$\leq \frac{1}{1-\gamma}\sqrt{\frac{\beta W^2 \log(A)}{T}} + \frac{56}{(1-\gamma)^2}\sqrt{\frac{C_{\mathrm{npg},\pi^\star}}{T}}.$$

Thus, to ensure that $\mathbb{E}\left[V^{\pi^\star} - V^{\widehat{\pi}}\right] \leq \varepsilon$, we set

$$T = O\left(\frac{1}{\varepsilon^2}\left(\frac{\beta W^2 \log(A)}{(1-\gamma)^2} + \frac{C_{\mathrm{npg},\pi^\star}}{(1-\gamma)^4}\right)\right).$$

This implies a total number of online samples, as

$$O\left(\frac{\log(2\max\{(W/T)^d, |\mathcal{F}|\}/\delta)}{\varepsilon^6 (1-\gamma)^8}\left(\beta W^2 \log(A) + \frac{1}{(1-\gamma^2)}C_{\mathrm{npg},\pi^\star}\right)^3\right).$$

Total number of offline samples used is same as above.

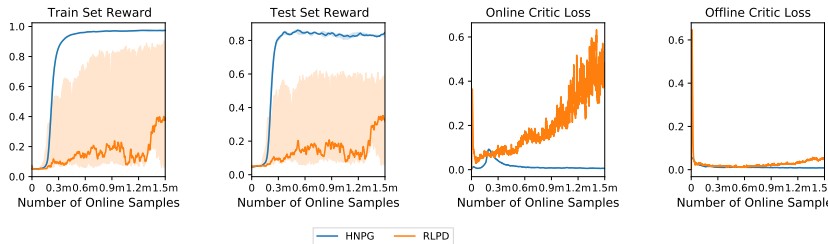

Figure 3: Comparison of the moving average learning and loss curves between HNPG and RLPD on an image-based Comblock with horizon 5. For train/test reward curves, median over 5 random seeds are reported and 20/80th quantile are shaded. For loss curves, one random run is chosen and reported. Online critic loss for HNPG refers to online MC regression square loss and offline critic loss for HNPG refers to offline TD loss. Both online and offline critic losses for RLPD are TD losses.

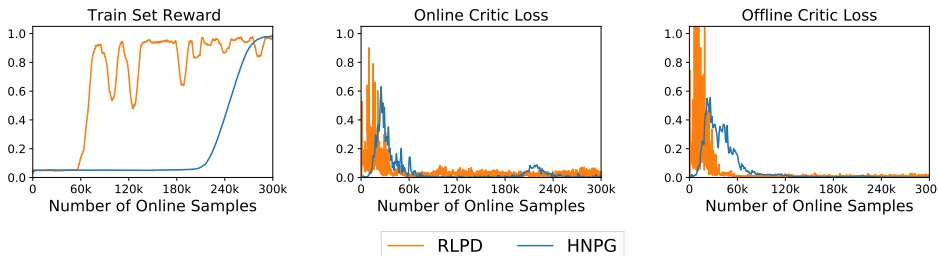

Figure 4: Comparison of the loss curves between HNPG and RLPD on a continuous Comblock with horizon 15.

# E  EXPERIMENT DETAILS OF COMBLOCK

## E.1  DETAILS OF COMBINATION LOCK

In a Comblock environment, each timestep has three latent states with the first two being good states and the last one being an absorbing state. Each latent state has 10 underlying actions. In good states, only one underlying action will lead to one of two good states of the next timestep with equal probability while the rest of the underlying actions will lead to the absorbing state of the next timestep. Once the agent gets to an absorbing state, any underlying action will lead to the absorbing state of the next timestep. Once the agent reaches one of the good states in the last timestep, it will receive an optimal reward of 1. When the agent goes from a good state to an absorbing state, it also has a 0.5 probability of receiving an anti-shaped reward of 0.1. Rewards for any other transitions are 0. To get the observation for each latent state, we concatenate one-hot representations of the latent state and horizon, add random noise $\mathcal{N}(0, 0.1)$ to each dimension, and finally multiply it with a Hadmard matrix.

## E.2  LOSS CURVES FOR COMBLOCK EXPERIMENTS

In Figure 4, we present the loss curve comparison for HNPG and RLPD on a continuous Comblock with horizon 5. Although RLPD enjoys a smaller sample complexity compared to HNPG, its learning curve is less stable. Similar to Figure 3, the online critic loss is more bumpy for RLPD since it optimizes TD loss and its bellman bootstrapping can be unstable.

## E.3  IMPLEMENTATION PSEUDOCODE

Following prior combination lock algorithms (Song et al., 2023; Zhang et al., 2022b), instead of a discounted setting policy evaluation, we adapt to the finite horizon setting and train separate Q-functions and policies for each timestep. We also incorporated some empirical recommendations from (Schulman et al., 2015) and the resulting practical algorithm is presented in Algorithm 4 and Algorithm 5.

---

**Algorithm 4** Practical Finite-Horizon HNPG

---

**Require:** Function class $\{\mathcal{F}_i\}_{i=1}^{H-1}$, PG iteration $T$, offline data $\nu$, Params $\lambda$, KL constraint $\max_{\text{KL}}$.
1: Initialize $f_0^0, \ldots, f_{H-1}^0 \in \mathcal{F}$, and $\theta_0^1, \ldots, \theta_{H-1}^1$.
2: **for** $t = 1, \ldots, T$ **do**
3:    $f_0^t, \ldots, f_{H-1}^t \in \mathcal{F}, \mathcal{D}_{\text{off}}, \mathcal{D}_{\text{on}} \leftarrow \text{FHPE}(\{\mathcal{F}_i\}_{i=1}^{H-1}, \nu, \{\pi_{\theta_i^t}\}_{i=0}^{H-1}, \lambda)$.
4:    **for** $h = 0, \ldots, H - 1$ **do**
5:       Let $\phi_h^t(s, a) = \nabla \log \pi_{\theta_h^t}(a|s)$ and $\bar{f}_h^t(s, a) = f_h^t(s, a) - \mathbb{E}_{a \sim \pi_{\theta_h^t}(s)}[f_h^t(s, a)]$.
6:       Use Conjugate Gradient to solve:

$$w_h^t \in \underset{w}{\operatorname{argmin}} \, \widehat{\mathbb{E}}_{\mathcal{D}_{\text{off}}^h} \left[ (w^\top \phi_h^t(s, a) - \bar{f}_h^t(s, a))^2 \right] + \lambda \widehat{\mathbb{E}}_{\mathcal{D}_{\text{on}}^h} \left[ (w^\top \phi_h^t(s, a) - \bar{f}_h^t(s, a))^2 \right].$$

(48)

7:       Get $\eta^t = \operatorname{argmax}_\eta \ell_{\text{LS}}(\eta, \max_{\text{KL}})$ according to (50) using line search.
8:       Update $\theta_h^{t+1} \leftarrow \theta_h^t + \eta^t w_h^t$.
9:    **end for**
10: **end for**
11: **Return** policy $\pi_{\theta_0^T}, \ldots, \pi_{\theta_{H-1}^T}$

---

**Algorithm 5** **F**inite-Horizon **H**ybrid **F**itted **P**olicy **E**valuation (FHPE)

---

**Require:** Policy $\pi_0, \ldots, \pi_{H-1}$, function class $\{\mathcal{F}_i\}_{i=0}^{H-1}$, offline distribution $\nu$, weight $\lambda$
1: Initialize $f_0, \ldots, f_{H-1} \in \mathcal{F}, f_H = 0$.
2: Sample $\mathcal{D}_{\text{on}} = \{\{(s_i, a_i, y_i = \widehat{Q}_i^\pi(s, a))\}\}_{i=0}^{H-1}$ of $m_{\text{on}}$ many on-policy samples using $\pi_0, \ldots, \pi_{H-1}$.
3: Sample $\mathcal{D}_{\text{off}} = \{\{(s_i, a_i, s_i', r_i)\}\}_{i=0}^{H-1}$ of $m_{\text{off}}$ many offline samples from $\nu$.
4: **for** $h = H - 1, \ldots, 0$ **do**
5:    Solve the square loss regression problem to compute:

$$f_h \leftarrow \underset{f \in \mathcal{F}_h}{\operatorname{argmin}} \, \widehat{\mathbb{E}}_{\mathcal{D}_{\text{off}}^h} (f(s, a) - r - f_{h+1}(s', \pi_{h+1}(s')))^2 + \lambda \widehat{\mathbb{E}}_{\mathcal{D}_{\text{on}}^h} (f(s, a) - y)^2. \quad (49)$$

6: **end for**
7: **Return** $f_0, \ldots, f_{H-1}$, and optionally $\mathcal{D}_{\text{off}}$ and $\mathcal{D}_{\text{on}}$.

---

In Algorithm 4, we define the line search objective $\ell_{\text{LS}}^t(\eta, \max_{\text{KL}})$, which take the following form:

$$\ell_{\text{LS}}^t(\eta, \max_{\text{KL}}) = L_{\theta_h^t}(\theta_h^t + \eta w_h^t) - \mathcal{X}\{\text{KL}(\theta_h^t, \theta_h^t + \eta w_h^t) \leq \max_{\text{KL}}\}, \quad (50)$$

where $L(\cdot)$ is the policy gradient objective, and $\mathcal{X}\{\cdot\} = 0$ if the statement is true and $\infty$ otherwise. In practice, this objective is solved using line search. For more details we refer the reader to the original paper (Schulman et al., 2015).

### E.4 HYPERPARAMETERS

We provide the hyperparameters of HNPG for both continuous Comblock and image-based continuous Comblock in Table 1. In addition, we provide the hyperparameters we tried for RLPD baseline for both Comblock settings in Table 2.

Table 1: Hyperparameters for HNPG in (image-based) continuous Comblock

|  | Value Considered | Final Value |
|---|---|---|
| GAE $\tau$ | {0.97, 0.9} | 0.97 |
| L-2 regularization rate | {0, 1e-3, 1e-2} | 0 |
| Maximum KL difference | {1e-1, 1e-2, 1e-3} | 1e-2 |
| Damping | {1e-1} | 1e-1 |
| Optimizer | {Adam, SGD} | Adam |
| Batch size | {500, 1000} | 1000 |
| Reweighting factor $\lambda$ | {0.1, 1, 10} | 1 |

Table 2: Hyperparameters for RLPD in (image-based) continuous Comblock

|  | Value Considered | Final Value |
|---|---|---|
| Discount $\gamma$ | {0.99} | 0.99 |
| Actor minimum standard deviation | {-10} | -10 |
| Actor maximum standard deviation | {2} | 2 |
| Initial temperature | {0.1} | 0.1 |
| Alpha Beta | {0.5} | 0.5 |
| Alpha Learning Rate | {1e-4} | 1e-4 |
| Actor learning rate | {1e-2, 1e-3} | 1e-3 |
| Critic learning rate | {1e-2, 1e-3} | 1e-3 |
| Critic soft update $\tau$ | {0.01, 0.02, 0.1} | 0.01 |
| Critic soft update frequency | {1, 2} | 2 |
| Optimizer | {Adam} | Adam |
| Number of updates per sample | {1, 10} | 1 |
| Batch size | {64, 128} | 128 |
| Buffer size | {1e5, 1e6} | 1e6 |

