# OpenReview forum: "Offline Data Enhanced On-Policy Policy Gradient with Provable Guarantees"
_ICLR.cc/2024/Conference — ICLR 2024 poster_

### Official Review · Reviewer_CW8P · 2023-10-30

**Soundness:** 3 good
**Presentation:** 3 good
**Contribution:** 3 good
**Rating:** 6
**Confidence:** 4

**Summary:**

The paper studied the hybrid RL setting where the agent has access to both online and offline dataset. A natural policy gradient based algorithm is proposed with provable guarantee on the sample complexity. The sample complexity bound showed that the approach, in theory, can achieve the best of both worlds. The paper further provides simulation studies on combinatory lock environment, where the proposed algorithm showed advantage over baseline methods.

**Strengths:**

The hybrid RL setting has a quickly growing literature and this paper by considering a natural policy gradient algorithm is a significant contribution to the literature. The algorithm has sound theoretical guarantee.

**Weaknesses:**

In general, the discussions on the theoretical results have to be more comprehensive and more coherent.

The paper mentioned in the abstract that the sample complexity bound is best-of-both-worlds. However, I did not see any discussion on the existing bound for pure online and pure offline bound. Many results in the offline literature, when a pessimism algorithm is applied, has a sample-complexity bound depending on the single-policy concentrability instead of the all-policy. it is clearly not the best in the world of pure-offline learning.

The NPG coverage condition and the Bellman error transfer coefficient do not seem to match. Could NPG coverage condition be weakened such that it reflects the dependence on the value function class? In the current form, if $\nu$ is generated by some policy on the environment, then $C_{off, \pi^e} \leq C_{jpg, \pi^e}$.

Could the authors comment on the technical contribution? By quickly going through the appendix, I don't see significant technical contributions except following the analysis in Song et al. (2022) and the previous natural policy gradient literature.

One can image that the algorithm performance should heavily depend on the coverability of the offline dataset. I suggest the authors test the effects of offline distribution $\nu$. Does the a better coverage provide better performance?

**Questions:**

The authors proposed a parameterized policy class. What is a sufficient condition on the policy class to ensure the same sample complexity bound in Theorem 1?

As indicated by the Theorem 1, the algorithm should benefits more from the offline dataset when Bellman completeness is not satisfied. Has this been observed in the simulation studies? It seems that the HNPG always has the dominate performance regardless of the Bellman completeness? Do the authors have any comments on this point? I believe this is critical regarding how the theoretical results help us understand the real-world performance.

---

> ### Author Response · Authors · 2023-11-19
>
> Thanks for your review! We would like to address the following concerns that have been raised.
>
> ### Our results depend on an all-policy concentration while recent offline rl literature only depends on single-policy concentrability.
> We would like to point out that our definition of $C_{off}$ is, in fact, ***single-policy*** concentration, instead of all-policy concentration. In particular, it is consistent with the form of concentration used by pessimism-based offline RL literature [1], where in their practical algorithm they also need the concentricity to hold for all $\mathcal{T}^\pi,\pi \in \{ \pi_1,..,\pi_T\}$ generated by the policy gradient algorithm. In principle, we could also write our assumptions in the same way, but since $\pi_1,...,\pi_T$ are all algorithm-dependent quantities, we simply replace it with $\max_\pi$. Furthermore, it can be shown that $C_{off}$ is weaker than the single-policy density ratio.
>
> [1] Xie, Tengyang, et al. "Bellman-consistent pessimism for offline reinforcement learning." Advances in neural information processing systems 34 (2021): 6683-6694.
>
> ### Could we weaken the assumption of $C_{NPG, \pi^e}$ to be dependent on some functional forms?
> Yes, we can replace the density ratio at least by a fraction of square error under the two distributions in the form of $\max_\pi \max_{f \in \mathcal{F}}\frac{E_{s,a \sim d^{\pi^e}}(f(s,a) - Q^\pi(s,a))^2}{E_{s,a \sim d^{\pi}}(f(s,a) - Q^\pi(s,a))^2}$. Indeed, Equation 22 in the appendix is the only place where we used the definition of $C_{npg}$ and we can relax the assumption to the minimum needed for that inequality to hold. We will add a discussion on this in the final version of the paper.
>
> ### What is the technical contribution of the paper?
> The main technical challenge of this paper is to achieve best-of-both-world type guarantees where we can still achieve meaningful results where either offline or online assumptions fail. This can be particularly meaningful in real-world problems where it is hard to verify the assumptions. To do this, we propose a novel HPE subroutine in Algorithm 2, where we optimize simultaneously for TD error and monte carlo regression loss and we need to carefully balance the statistical error term from the TD error and monte carlo regression such that the error of the other term does not blow up when the realizability of one term is not met. In fact, the TD error and the MC regression error interleave with each other and our analysis uses one to supervise the other (e.g., MC regression error is used to bound the Bellman residual under the offline distribution). The entirety of Appendix B is dedicated to this proof.
>
> ### Does a better coverage of the offline distribution provide a better performance?
> In our experiments, we follow a similar setup to [1] where the offline distribution is generated by an epsilon greedy policy (on top of the optimal policy) to ensure both sufficient coverage and the average performance of the offline data is around 3% of the optimal policy. The choice of offline distribution is carefully chosen such that $C_{off,\pi_e}$ is bounded while naive methods such as behavior cloning is only as good as the epsilon-greedy policy.
>
> [1] Song, Yuda, et al. "Hybrid rl: Using both offline and online data can make rl efficient." arXiv preprint arXiv:2210.06718 (2022).
>
> ### What is a sufficient condition on the policy class to ensure the same sample complexity bound in Theorem 1?
> The full assumptions, update rules, and discussions on parameterized policy class are presented in Appendix D. We will move it to the main body for the final version of the paper.
>
> ### How does the simulation reflect the benefit of HNPG when Bellman completeness is not satisfied? It seems that the HNPG always has the dominant performance regardless of the Bellman completeness?
> Although it is hard to tell from Figure 2, a comparison between the learning curves and loss curves in Figure 3 and Figure 4 in the appendix shows that HNPG is actually more at an advantage in the case of image-based comblock without Bellman completeness. In Figure 4, we can see that with Bellman completeness, in the short horizon case, RLPD can actually learn faster if tuned properly. However, by examining the loss curve we can find that the online TD error has a much larger fluctuation compared to the online Monte Carlo regression loss, potentially due to the instability of optimization in practice caused by bootstrapping in TD-style algorithms. This could explain why even with Bellman completeness, HNPG dominates the performance in the long horizon case.

---

> > ### Comment · Reviewer_CW8P · 2023-11-20
> >
> > Thanks for the answers, it helps clarify some concerns. I do not have further questions.

---

### Official Review · Reviewer_mti6 · 2023-11-01

**Soundness:** 4 excellent
**Presentation:** 4 excellent
**Contribution:** 4 excellent
**Rating:** 8
**Confidence:** 3

**Summary:**

This paper proposed a hybrid RL method based on NPG that uses both on-policy data and offline data. The algorithm enjoys theoretical guarantees for the best of both the offline RL setting and on-policy RL setting. Empirical performance also shows that the proposed hybrid RL algorithm outperforms the existing algorithms in challenging scenarios.

**Strengths:**

Clarify: the paper is well written. The motivation is justified clearly. The algorithm design and theoretical guarantees are also explained clearly.

Significance and quality: the results are quite significant since it is able to achieve the best of the both world theoretical guarantees, with empirical performance improvements in challenging scenarios too.

**Weaknesses:**

I don't see major weaknesses. Just one question about Figure 2: it is hard to differentiate the curves generated by different approaches, thus difficult to see the message from this figure. I suggest the authors improve the presentation of Figure 2.

**Questions:**

See above.

---

> ### Author Response · Authors · 2023-11-19
> **Rebuttal**
>
> Thanks for your review! We would definitely improve the presentation of Figure 2 to make it clearer to tell the difference between two approaches.

---

### Official Review · Reviewer_HGgg · 2023-11-01

**Soundness:** 2 fair
**Presentation:** 2 fair
**Contribution:** 2 fair
**Rating:** 6
**Confidence:** 4

**Summary:**

This paper proposes a new hybrid reinforcement learning (RL) algorithm called Hybrid Actor-Critic (HAC) that combines on-policy and off-policy learning. The proposed algorithm uses compatible function approximation and a hybrid loss using both online and offline data to achieve better theoretical guarantees. The authors present a theoretical analysis to show that it has both offline and on-policy performance guarantees with a small Bellman error and still has the on-policy performance gurantee when the Bellman error is large. In practice, the algorithm is tested on rich-observation and exploration-challenging environments and is shown to outperform hybrid RL baselines like RLPD.

**Strengths:**

1. Theoretical analysis: The paper provides a thorough theoretical analysis of the proposed algorithm, demonstrating that it achieves nice theoretical guarantees when offline RL-specific assumptions hold, while maintaining the guarantees of on-policy natural policy gradient (NPG) regardless of the validity of the offline RL assumptions.
2. Empirical results: The authors demonstrate the effectiveness of their approach on challenging rich-observation environments, outperforming state-of-the-art hybrid RL baselines. This showcases the practical benefits of combining on-policy and off-policy learning.

**Weaknesses:**

1. The novelty of the proposed method is limited. The algorithm is a combination of natural actor-critic and hybrid-RL [1], and it needs to be clarified what is the technical challenge in combining them.

2. I am also concerned with the claim that the proposed method achieves the best of both worlds. While it is nice to have guarantees without the Bellman completeness assumption, other strong assumptions are still required. The realizability for any $Q^\pi$ is a strong assumption, and the value-based method usually only requires the realizability of $Q^*$. More importantly, the assumption on NPG coverage is also strong since it is an all-policy coverage assumption. Offline RL only needs single-policy coverage assumption as in Definition 3, and it can be proved that $C_{off,\pi^e}^2 \leq C_{npg,\pi^e}$ [1,2]. This invalidates the claim that the proposed method achieves the best of both worlds since when the offline assumption does not hold, the online guarantee is lost, too. I expect an exploration mechanism so that we can still achieve low regret when the offline data has insufficient coverage.

[1] Song, Yuda, et al. "Hybrid rl: Using both offline and online data can make rl efficient." arXiv preprint arXiv:2210.06718 (2022).

[2] Xie, Tengyang, et al. "Bellman-consistent pessimism for offline reinforcement learning." Advances in neural information processing systems 34 (2021): 6683-6694.

**Questions:**

1. What is the technical challenge in combining natural actor critic and hybrid RL?

2. When the offline guarantee does not hold, how can we still retain the on-policy guarantee?

3. Can you provide more insights on the superior performance of the proposed method on the comblock environment?

---

> ### Author Response · Authors · 2023-11-19
> **Rebuttal**
>
> Thanks for your review! We would like to address the following concerns that have been raised.
>
> ### What is the technical challenge in combining Hybrid RL with natural actor critic?
> The main technical challenge of this paper is to achieve best-of-both-world type guarantees where we can still achieve meaningful results where either offline or online assumptions fail. This can be particularly meaningful in real-world problems where it is hard to verify the assumptions. To do this, we propose a novel HPE subroutine in Algorithm 2, where we optimize simultaneously for TD error and monte carlo regression loss and we need to carefully balance the statistical error term from the TD error and monte carlo regression such that the error of the other term does not blow up when the realizability of one term is not met. In fact, the TD error and the MC regression error interleave with each other and our analysis uses one to supervise the other (e.g., MC regression error is used to bound the Bellman residual under the offline distribution). The entirety of Appendix B is dedicated to this proof.
>
> We emphasize that, to the best of our knowledge, and despite being simple, our hybrid PE algorithm 2 is new. Prior work either just focuses on pure on-policy algorithms or pure offline TD-style algorithms. There is no provable work that attempts to combine them together. By combining these two components together into a single objective in a principled manner, we managed to get the best of both worlds style results.
>
> ### The assumption for online RL such as realizability of Q^{\pi} and the definition of $C_{NPG}$ is too strong and invalidates the claim of best of both worlds.
> We would like to first note that $Q^{\pi}$ realizability is a very standard assumption in the Policy Gradient literature. For example, prior works (including [1,2,3]) all use this Q^\pi realizable assumption (indeed, assumptions in [1,3] are stronger than Q^\pi being realizable).
>
> Furthermore, our definition of $C_{NPG}$ is only ***single-policy*** concentration with respect to a single comparator policy $\pi_e$ and it can be shown that our definition of $C_{NPG}$ is weaker than the single-policy density ratio $\|\frac{d^{\pi^e}}{\nu}\|_{\infty}$ where $\nu$ is the reset distribution. The notion of a good reset distribution is also a standard assumption in PG literature [4,5]. In particular, Page 124 Figure 0.1 of [5] shows that without a good reset distribution, PG methods can easily get stuck in a local optimum.
>
> There are indeed two cases where online guarantees hold while offline guarantees fail. Firstly, when the offline distribution is not admissible (generated by some policy) we can no longer show $C_{off, \pi_e}^2 < C_{npg, \pi_e}$, and it is possible that $C_{off, \pi_e}^2$ is unbounded while $C_{npg, \pi_e}$ is bounded. Secondly, and more importantly as shown in our simulation experiments in Section 6, when the offline distribution is admissible but the Bellman completeness assumption does not hold, the online guarantee of our algorithm is still meaningful.
>
>
> [1] Xie, Tengyang, et al. "Bellman-consistent pessimism for offline reinforcement learning." Advances in neural information processing systems 34 (2021): 6683-6694.
>
> [2] Weisz, Gellert, et al. Online RL in Linearly \$q\textasciicircum\pi\$-Realizable MDPs Is as Easy as in Linear MDPs If You Learn What to Ignore, In Advances in Neural Information Processing Systems. MIT
>
> [3] Qinghua Liu, Gellért Weisz, András György, Chi Jin, & Csaba Szepesvári. (2023). Optimistic Natural Policy Gradient: a Simple Efficient Policy Optimization Framework for Online RL.
>
> [4] Kakade, S. (2001). A Natural Policy Gradient. In Advances in Neural Information Processing Systems. MIT Press.
>
> [5] Agarwal, A, Jiang N, Kakade, S, Sun, W. (2022). Reinforcement Learning, Theory and Algorithms.
>
>
> ### Intuition of the superior performance of HNPG in the comblock environment.
> Compared with pure online RL such as TRPO, HNPG is better in the use of offline data to help explorations in hard exploration environments such as comblock. Comparing state-of-the-art hybrid RL approaches such as RLPD, they rely heavily on Bellman completeness by doing bootstrapping for both online and offline data. In HNPG, the dependence on Bellman completeness is relaxed by doing Monte Carlo regression for online data thus working better for the real image observations where Bellman completeness does not necessarily hold.

---

> > ### Comment · Area_Chair_oCyr · 2023-11-20
> > **author reviewer discussion is ending soon**
> >
> > Dear HGgg,
> >
> > The author reviewer discussion period is ending soon this Wed. Does the author response clear your concerns regrading novelty and strong assumptions? Are there any other outstanding questions that you would like the authors to address?
> >
> > Thanks again for your service to the community.
> >
> > Best,
> > AC

---

> > ### Comment · Reviewer_HGgg · 2023-11-22
> >
> > Thank you for the response. The response resolved my major concerns, especially the "best of both worlds" claim, and I increased my score to 6. However, I would like to clarify that $C_{npg,\pi^e}$ in definition 2 is *not* a single-policy converage coefficient since it requires all policies $\pi$ to satisfy the condition $|d^{\pi^e}/d^\pi|\_\infty \leq C_{npg,\pi^e}$. On the contrary, the offline coefficient only requires the coverage of $\nu$, which can be much smaller than the uniform coverage coefficient. Another reason the assumption on finite $C_{npg,\pi^e}$ is strong is due to the infinite norm on the state-action distribution. It is unlikely that all policies can cover the whole state-action space, especially for continuous state spaces. It is an interesting direction to eliminate the dependence on the uniform coverage coefficient by introducing proper exploration like [1].
> >
> > [1] Cai, Qi, et al. "Provably efficient exploration in policy optimization." International Conference on Machine Learning. PMLR, 2020.

---

### Official Review · Reviewer_M4D3 · 2023-11-03

**Soundness:** 3 good
**Presentation:** 4 excellent
**Contribution:** 3 good
**Rating:** 8
**Confidence:** 3

**Summary:**

This paper focuses on hybrid RL, where the agent has offline data and is able to interact with the environment simultaneously. The novel algorithms introduced in this paper integrate the TD error from offline data and the estimation error coming from online datasets into the policy evaluation loss. These approaches distinguish itself from the prior works that directly merge offline and online data, followed by applying an off-policy method. The theoretical result shows that the optimality gap of the proposed method can be upper bounded without Bellman Completeness or with Bellman Completeness and offline data coverage guarantee. The experiments are conducted in a continuous Comblock environment, which necessitates accurate multi-step actions to attain the final optimal reward. The results reveal that this method outperforms TRPO, a purely on-policy method, as well as RLPD, a hybrid off-policy actor-critic method.

**Strengths:**

- The proposed algorithms in the hybrid RL setting are interesting and novel.
- The theoretical guarantee established for the two distinct cases of Bellman Completeness offers significant insights. To my knowledge, this paper is the first that provides an algorithm with best-of-both-worlds type guarantees for hybrid RL (so that the assumption about Bellman Completeness can be relaxed).
- The proposed algorithms empirically outperform other hybrid RL baseline like RLPD in hard exploration problems (continuous and image-based Comblock).
- The paper is very well-written and easy to follow.

**Weaknesses:**

I did not spot any particular major weakness in this paper. That said, there are a few places that could be further improved:
- In Algorithm 3, HNPG solves the squared loss regression with a linear critic. The requirement of a linear critic is probably mainly for theoretical analysis and could be relaxed to more general function classes (e.g., neural function approximation). A further discussion (possibly with some experiments) could make the proposed algorithm more impactful.
- Regarding Definition 3, it is a bit difficult to get a sense of how large the transfer coefficient could be (as it is the maximum taken over all stationary policies and the whole function class) and under what condition of $\nu$ would $C_{off}$ be bounded or small.
- As the proposed HNPG focuses on the hybrid RL problems, readers would probably expect at least a bit of experimental comparison on those more mainstream RL benchmarks like D4RL (as done in several recent hybrid RL papers, e.g., RLPD and Cal-QL) or hard exploration problems in Atari, e.g., Montezuma's Revenge as in (Song et al., 2022).

**Questions:**

Some detailed questions:
- The concept and the terminology of “NPG coverage” in Definition 2 could be further elaborated on. Specifically, Definition 2 is defined w.r.t. some comparator policy, which is not necessarily related to NPG.
- To calculate the loss in Eq. (1), one would need to obtain an unbiased estimate $y$, which could be obtained via Monte Carlo sampling. On the other hand, it would probably be better to get $y$ with the help of bootstrapping. Would this still preserve a similar sub-optimality guarantee as in Theorem 1?
- Regarding Definition 2, how to ensure a finite $C_npg$ without the assumption on the positivity of the reset distribution?

---

> ### Author Response · Authors · 2023-11-19
> **Rebuttal**
>
> Thanks for your review! We would like to address the following concerns that have been raised.
>
> ### The requirement of a linear critic in Algorithm 3.
> We would like to clarify that the linear critic in Algorithm 3 is practical for the following two reasons: 1) the feature used in Algorithm 3 is $\nabla \ln \pi(a|s)$ which is equal to $\nabla \pi(a|s)/\pi(a|s)$ where $\nabla \pi(a|s)$ is the neural tangent kernel, i.e., define $\phi(s,a) := \nabla \pi(a|s)$, the kernel $k(z,z’) = \phi(z)^T \phi(z’)$ is called a neural tangent kernel ([1] and Lemma 8.1.1 of [2]). Therefore, the linear regression in Algorithm 3 is actually neural tangent kernel linear regression, and the neural tangent kernel has been shown to be expressive. 2) Practical implementations of NPG like TRPO uses fisher information matrix to precondition the gradient and it can shown that it is equivalent to first do regression with $\nabla \ln \pi(a|s)$ feature.
>
>
> ### How big is $C_{off}$ in Definition 3?
> We would like to note that $C_{off}$ is only a single-policy concentrability instead of an all-policy concentrability, and it is the same as many practical algorithms in the offline RL literature that uses NPG analysis for offline RL. In particular, it is consistent with the form of concentration used by pessimism-based offline rl literature [3], where in their practical algorithm they also need the concentrability to hold for all $\mathcal{T}^\pi,\pi \in \{ \pi_1,..,\pi_T\}$ generated by the policy gradient algorithm. In principle, we could also write our assumption in the same way, but since $\pi_1,...,\pi_T$ are all algorithm-dependent quantities, we simply replace it by $\max_\pi$. Furthermore, it can be shown that it is smaller than the single-policy density ratio $\|\frac{d^{\pi^*}}{\nu}\|_{\infty}$ where $\nu$ is the reset distribution.
>
>
> ### The definition of NPG with respect to some comparator policy?
> Our definition of NPG only requires coverage of a single policy instead of requiring coverage of all policies (as commonly used in the NPG literature such as [4,5]). It can be shown that our definition of $C_{NPG}=\max_\pi |\frac{d^{\pi^*}}{d^\pi}|_{\infty}$
>
> is weaker than the single-policy density ratio of the reset distribution $\|\frac{d^{\pi^*}}{\nu}\|_{\infty}$ except a factor of $\frac{1}{1-\gamma}$.
>
> We also note that positivity of reset distribution, while being sufficient is not necessary for bounded $C_{NPG}$. In cases where we do not have positivity of the reset distribution, $C_{NPG}$ could be bounded when the MDP dynamics is favorable. As an example, a typical case where $C_{NPG}$ holds is Ergodic MDP (for example, see Assumption 2 from [6]) which implies that the probability of any policy to visit any reachable state is strictly positive.
>
>
> ### In equation (1), can we obtain y with bootstrapping?
> Using bootstrapping to get y typically requires the assumption of Bellman completeness (e.g., algorithms like TD which use bootstrapping often require very specific assumptions to guarantee to succeed), which is not needed in standard NPG analysis. In order to achieve online RL guarantees when the assumption of Bellman completeness does not hold, we choose to use Monte Carlo estimation for y.
>
> [1] Jacot, A., Gabriel, F., & Hongler, C. (2018). Neural Tangent Kernel: Convergence and Generalization in Neural Networks. In Advances in Neural Information Processing Systems. Curran Associates, Inc..
>
> [2] Raman, A., et al. “The Theory of Deep Learning”
>
> [3] Xie, Tengyang, et al. "Bellman-consistent pessimism for offline reinforcement learning." Advances in neural information processing systems 34 (2021): 6683-6694.
>
> [4] Kakade, S. (2001). A Natural Policy Gradient. In Advances in Neural Information Processing Systems. MIT Press.
>
> [5] Agarwal, A, et al. (2022). Reinforcement Learning, Theory and Algorithms.
>
> [6] Fabien Pesquerel, & Odalric-Ambrym Maillard (2022). IMED-RL: Regret optimal learning of ergodic Markov decision processes. In Advances in Neural Information Processing Systems.

---

> ### Comment · Reviewer_M4D3 · 2023-11-22
>
> Thanks for the detailed response. All of my questions have been addressed.

---

### Meta-Review · Area_Chair_oCyr · 2023-12-05

**Metareview:**

This paper proposes a novel approach to use both online and offline data in actor critic algorithms. In particular, online data and offline data factor into different loss terms in updating the critic. A theoretical analysis is given without requiring Bellman completeness and the proposed approach outperforms both pure online and hybrid methods in the challenging Comblock tasks.

Strength: Overall this paper is a solid work and is well written. The critic loss combines offline and online data in a novel way, which makes it possible to conduct analysis without requiring Bellman completeness. Performance boost is observed in the challenging Comblock task with image as input.

Weakness: It would be better to provide numerical verification of the assumptions in some small tasks to provide better understanding of the assumptions. It would also be more informative to provide empirical comparison in more standard RL benchmarks, e.g., D4RL as used by RLPD and Montezuma's Revenge as used by HyQ, to better place the proposed method in the literature.

**Justification For Why Not Higher Score:**

Lack of extensive empirical study in standard RL benchmarks. Lack of small scale experiments to numerically verify the assumptions.

**Justification For Why Not Lower Score:**

All reviewers agree this is a solid work.

---

### Decision · Program_Chairs · 2024-01-16

Accept (poster)